# Denoising Task Difficulty-based Curriculum for Training Diffusion Models

**Jin-Young Kim**[*][†]    **Hyojun Go**[*]    **Soonwoo Kwon**[*]    **Hyun-Gyoon kim**[1][†]
Ajou University[1]
{seago0828, gohyojun15, swkwon.john}@gmail.com, hyungyoonkim@ajou.ac.kr

## Abstract

Diffusion-based generative models have emerged as powerful tools in the realm of generative modeling. Despite extensive research on denoising across various timesteps and noise levels, a conflict persists regarding the relative difficulties of the denoising tasks. While various studies argue that lower timesteps present more challenging tasks, others contend that higher timesteps are more difficult. To address this conflict, our study undertakes a comprehensive examination of task difficulties, focusing on convergence behavior and changes in relative entropy between consecutive probability distributions across timesteps. Our observational study reveals that denoising at earlier timesteps poses challenges characterized by slower convergence and higher relative entropy, indicating increased task difficulty at these lower timesteps. Building on these observations, we introduce an easy-to-hard learning scheme, drawing from curriculum learning, to enhance the training process of diffusion models. By organizing timesteps or noise levels into clusters and training models with ascending orders of difficulty, we facilitate an order-aware training regime, progressing from easier to harder denoising tasks, thereby deviating from the conventional approach of training diffusion models simultaneously across all timesteps. Our approach leads to improved performance and faster convergence by leveraging benefits of curriculum learning, while maintaining orthogonality with existing improvements in diffusion training techniques. We validate these advantages through comprehensive experiments in image generation tasks, including unconditional, class-conditional, and text-to-image generation.

## 1 Introduction

Diffusion-based generative models (Ho et al., 2020; Sohl-Dickstein et al., 2015; Song et al., 2021) have achieved significant advancements in the realm of generative tasks, demonstrating notable success across various fields such as image (Dhariwal & Nichol, 2021), video (Ho et al., 2022; Harvey et al., 2022), and 3D (Woo et al., 2023; Liu et al., 2023b) generation. Specifically, their exceptional adaptability and promising performance in diverse image generation contexts, such as unconditional (Karras et al., 2022; Nichol & Dhariwal, 2021), class-conditional (Dhariwal & Nichol, 2021), and text-conditional scenarios (Balaji et al., 2022; Ramesh et al., 2022), demonstrate their significant impact. Such achievements have led to a growing interest in further deepening the analysis and enhancing diffusion models.

Diffusion models (Ho et al., 2020; Song et al., 2021) are designed to reverse the corruption of the data through the learning process at different noise levels and over multiple timesteps. Recent works have delved into the learning of diffusion models across various noise levels and timesteps, revealing different stages of diffusion models. For example, Choi *et al.* (Choi et al., 2022) observe that when a diffusion model performs a denoising task from large to small timestep, it first generates coarse features, then gradually generates perceptually rich content, and later refines the details. Similar observation is also identified in text-to-image diffusion models (Balaji et al., 2022). Besides this aspect, various studies have further explored the learning of diffusion models across timesteps and noise levels, elucidating their transition from denoising to generative functionalities (Deja et al., 2022), modular attributes (Yue et al., 2024), frequency characteristics (Yang et al., 2023b; Lee et al., 2023), trajectories (Pan et al., 2024), affinity (Go et al., 2023a), and variations of targets (Xu et al., 2023).

---

[*]Co-first author    [†]Corresponding author

These observations have not only deepened understanding of diffusion models but have also directly contributed to improvement in diffusion models. Specifically, these insights are incorporated into their method design in various works, including loss functions (Hang et al., 2023; Xu et al., 2023), architectures (Lee et al., 2023; Balaji et al., 2022), accelerated sampling (Pan et al., 2024), representations (Yue et al., 2024), and guidance (Go et al., 2023b). Given the tangible benefits already realized from such studies, further in-depth analysis of diffusion models across timesteps and noise levels is crucial for uncovering insights and achieving unprecedented advancements in their capabilities.

In this paper, to enrich the current understanding across various timesteps and noise levels, we investigate under-explored areas within diffusion models focusing on the *task difficulties* of denoising. Regarding denoising task difficulties, previous works speculate that denoising tasks across timesteps have different difficulties (Li et al., 2023; Balaji et al., 2022), yet a detailed exploration of these variances remains sparse. Moreover, there exists a notable discrepancy among studies, with works identifying larger timesteps as more difficult (Ho et al., 2020; Hang et al., 2023), while others argue that smaller timesteps pose greater difficulties (Karras et al., 2022; Dockhorn et al., 2021; Kim et al., 2022). The discrepancy in difficulty across timesteps not only impedes the accurate interpretation of previous studies but also hinders the development of sophisticated training methods that properly utilize the timestep-wise variation in difficulty.

In this regard, we first analyze task difficulties in two aspects to resolve these conflicts: 1) convergence properties in the learning of denoising tasks at each timestep, and 2) the change in relative entropy between consecutive probability distributions over timesteps. In the first aspect, our analysis reveals distinct convergence behaviors across timesteps, demonstrating that models trained on larger timesteps exhibit faster convergence. In the second aspect, we also observe a decrease in relative entropy as we progress to later timesteps. By integrating these, we confirm that denoising tasks at earlier timesteps are more difficult, indicated by slower convergence speeds and greater changes in relative entropy.

Furthermore, building on these observations, we integrate an easy-to-hard learning scheme, a concept well-established in the curriculum learning literature (Hacohen & Weinshall, 2019; Kong et al., 2021; Chang et al., 2021; Wang et al., 2020; Pentina et al., 2015), into the training process of diffusion models. Specifically, we organize timesteps or noise levels into clusters and train the diffusion models with ascending levels of difficulty, moving from clusters categorized by higher to lower timesteps. After this curriculum process, models simultaneously learn whole timesteps as standard diffusion training (Ho et al., 2020; Song et al., 2021; Ho & Salimans, 2022) to reach the convergence point. Unlike conventional approaches where diffusion models are trained simultaneously across all timesteps, our method distinguishes itself by incorporating a sequential, order-aware training regime, reflecting an intended progression from easier to harder denoising tasks.

Building upon this foundation, our curricular approach offers several notable advantages: **1) Improved Performance** and **2) Faster Convergence:** By leveraging the inherent benefits of curriculum learning, our method significantly enhances the quality of generation and the speed of convergence. **3) Orthogonality with Existing Improvements:** Our approach is inherently model-agnostic, ensuring broad applicability across various diffusion models. Additionally, it can be integrated with advanced diffusion training techniques, such as loss weighting (Choi et al., 2022; Hang et al., 2023; Go et al., 2023a; Karras et al., 2023).

Finally, we empirically validate the advantages of our method by conducting comprehensive experiments across a variety of image-generation tasks. These include unconditional generation, class-conditional generation, and text-to-image generation, utilizing datasets such as FFHQ (Karras et al., 2019), ImageNet (Deng et al., 2009), and MS-COCO (Lin et al., 2014). By integrating our curriculum learning strategy into architectures— DiT (Peebles & Xie, 2022), EDM (Karras et al., 2022), and SiT (Ma et al., 2024)—we demonstrate the efficacy of our approach in enhancing performance, accelerating convergence speed, and maintaining compatibility with existing techniques.

## 2 RELATED WORKS

### 2.1 DIFFUSION MODELS

Diffusion models (Ho et al., 2020; Sohl-Dickstein et al., 2015; Song et al., 2021) are a group of generative models that create samples by utilizing a learned denoising process to noise. Several works have focused on improving diffusion models in various aspects, including model architectures (Park et al., 2024b; Dhariwal & Nichol, 2021; Park et al., 2024a), sampling speed (Song et al., 2020; Lu

et al., 2022; Liu et al., 2023a), training objectives (Hang et al., 2023; Choi et al., 2022; Go et al., 2023a; Kingma & Gao, 2023; Ma et al., 2024). These endeavors often involve investigating what diffusion models learn by dividing its process, aiming to enhance the performance of diffusion models. P2 (Choi et al., 2022) under-weights training loss functions at the clean-up stage from their observation that diffusion models learn coarse, perceptual, and removing noises at large, medium, and small timesteps. Ediff-I (Balaji et al., 2022) observes that earlier sampling parts rely on conditions for generation, whereas later parts ignore the conditions. They employ multiple denoisers to address the diverse characteristics of tasks associated with different parts of the sampling process. Moreover, various works have also investigated these aspects related to timesteps (Deja et al., 2022; Yue et al., 2024; Yang et al., 2023b; Lee et al., 2023; Pan et al., 2024; Go et al., 2023a; Xu et al., 2023) (detailed illustrations can be found in Appendix A). While our study aligns with the above works, we analyze the under-explored aspect of denoising task difficulty. Furthermore, we leverage these observations to propose a curriculum learning approach.

## 2.2 DENOISING DIFFICULTIES ON DIFFUSION MODELS

Difficulties in denoising tasks in diffusion have been referred to by various works, but this aspect is not deeply explored. Several studies hypothesize that denoising tasks in diffusion encompass diverse difficulties (Li et al., 2023; Balaji et al., 2022), and there have been conflicts regarding these difficulties between previous works.

Certain studies consider denoising at larger noise levels and timesteps to be more difficult (Ho et al., 2020; Hang et al., 2023), the focus is on the challenges associated with reconstructing data from substantial noise. For instance, Hang *et al.* (Hang et al., 2023) articulate that while smaller timesteps (approaching zero) may require straightforward reconstructions, such strategies become less effective at higher noise levels or in larger timesteps. Similarly, Ho *et al.* (Ho et al., 2020) elucidate that their approach de-emphasizes loss terms at smaller timesteps to prioritize learning on the more challenging tasks at larger timesteps, thereby enhancing sample quality. Conversely, other studies argue that earlier timesteps or lower noise levels also present significant challenges. Karras *et al.* Karras et al. (2022) suggest that detecting noise at low levels is challenging due to its minimal presence. Also, Kim *et al.* (Kim et al., 2022) illustrate the increasing difficulty and high variance in score estimation as timesteps approach zero, disturbing stable training of models. In line with these observations, Dockhorn *et al.* (Dockhorn et al., 2021) build upon insights of (Kim et al., 2022), acknowledging the complexities at near zero timesteps, where score becomes highly complex and potentially unbounded.

In this work, we aim to resolve this conflict through an in-depth analysis of convergence properties and changes in relative entropy between consecutive probability distributions across timesteps.

## 2.3 CURRICULUM LEARNING

Curriculum learning (Bengio et al., 2009; Hacohen & Weinshall, 2019; Kong et al., 2021), inspired by human learning patterns, is a method of training models in a structured order, starting with easier tasks (Pentina et al., 2015) or examples (Bengio et al., 2009) and gradually increasing difficulty. As pointed out by (Bengio et al., 2009), curriculum learning formulation can be viewed as a continuation method (Allgower & Georg, 2003), which starts from a smoother objective and gradually transformed into a less smooth version until it reaches the original objective function. Through this foundation, various works have achieved improved performance and faster convergence compared to standard training based on random mini-batches sampled uniformly (Hacohen & Weinshall, 2019; Kong et al., 2021; Chang et al., 2021; Wang et al., 2020).

Curriculum learning primarily comprises two components: a curriculum scoring function, measuring the difficulty of tasks or examples, and a pacing function, modulating the speed of the curriculum progress. Regarding a curriculum score function, early studies have utilized human intuition for measuring difficulty, such as the complexity of geometric shapes in images (Bengio et al., 2009) or the length of sequences (Spitkovsky et al., 2010). Recently, various works employ models to measure difficulty, including confidence of pre-trained models (Hacohen & Weinshall, 2019) and the loss of the current models (Kong et al., 2021). For the pacing function, a predefined pacing function has been employed, which involves training using a predetermined curriculum progression (Hacohen & Weinshall, 2019; Wu et al., 2020). There are various forms of this and they can be generally represented as a function of training iteration (Hacohen & Weinshall, 2019; Wu et al., 2020). Contrary to this, there have been proposals for pacing techniques dynamically adjusting based on the loss or performance of the current model during training (Kumar et al., 2010; Jiang et al., 2014).

In the diffusion model literature, curriculum learning has been utilized to organize the order of training data types based on prior knowledge of targeted generation tasks (Tang et al., 2023; Yang et al., 2023a). Tang *et al.* (Tang et al., 2023) sequentially train video diffusion models with lower resolution and FPS datasets before progressing to higher resolution and FPS datasets. Similarly, Yang *et al.* (Yang et al., 2023a) order text-to-sound generation data based on the number of events in audio clips, training diffusion models from lower to higher events datasets. In contrast, our method explores the nature of denoising task difficulty in diffusion models and proposes a curriculum learning approach that progresses from easy to hard timesteps, deviating from the standard simultaneous training of all timesteps. Also, while consistency models (Song et al., 2023; Song & Dhariwal, 2023) adopt a curriculum approach to discretizing noise levels, progressively increasing the discretization steps of noise levels during training, we have distinct by exploring which noise level should be learned first and investigating the difficulty of denoising at each noise level.

## 3 PRELIMINARIES

In this section, we provide the necessary background on diffusion models (Ho et al., 2020; Sohl-Dickstein et al., 2015; Song et al., 2021). Let $\boldsymbol{x}_0 \in \mathbb{R}^d$ be a sample from the data distribution $p_0(\boldsymbol{x})$. The forward process of diffusion models transforms data $\boldsymbol{x}_0$ to latent $\boldsymbol{x}_{t \in [0,T]}$ by iteratively adding Gaussian noise. This can be formulated as a stochastic differential equation (SDE) (Song et al., 2021) as $\mathrm{d}\boldsymbol{x}_t = f(t)\boldsymbol{x}_t\mathrm{d}t + g(t)\mathrm{d}\boldsymbol{w}_t$, where $f(t)$ and $g(t)$ are drift and diffusion coefficients, and $\boldsymbol{w}_t$ is the standard Wiener process. The Gaussian transition kernel of this SDE is formulated as:

$$p_{0t}(\boldsymbol{x}_t|\boldsymbol{x}_0) = \mathcal{N}(\boldsymbol{x}_t; s_t\boldsymbol{x}_0, s_t^2\sigma_t^2\mathbf{I}), \quad s_t = \exp\left(\int_0^t f(\xi)\mathrm{d}\xi\right), \quad \sigma_t = \sqrt{\int_0^t \frac{g(\xi)^2}{s_\xi^2}\mathrm{d}\xi}. \quad (1)$$

For generation, diffusion models aim to learn the corresponding reverse SDE represented as:

$$\mathrm{d}\boldsymbol{x}_t = \left[f(t)\boldsymbol{x}_t - g^2(t)\nabla \log p_t(\boldsymbol{x}_t)\right]\mathrm{d}\bar{t} + g(t)\mathrm{d}\bar{\boldsymbol{w}}_t, \quad (2)$$

where $\bar{\boldsymbol{w}}_t$ and $\mathrm{d}\bar{t}$ denote the reverse-time Wiener process and the infinitesimal reverse-time, respectively, with the actual data score $\nabla \log p_t(\boldsymbol{x}_t)$. In most cases, a neural network $\epsilon_\theta$ having parameter $\theta$ is utilized to approximate this score function by learning the denoising tasks for each timestep $t$ from score matching loss $\mathcal{L}$ (Song & Ermon, 2019):

$$\mathcal{L} = \frac{1}{2}\int_0^T \mathcal{L}_t\mathrm{d}t, \quad \mathcal{L}_t = \omega(t)\mathbb{E}_{\boldsymbol{x}_t \sim p_{0t}(\boldsymbol{x}_t|\boldsymbol{x}_0),\boldsymbol{x_0} \sim p_0}\left[||\epsilon_\theta(\boldsymbol{x}_t, t) - \nabla \log p_{0t}(\boldsymbol{x}_t|\boldsymbol{x}_0)||_2^2\right], \quad (3)$$

where $\omega(t)$ is loss weights for $t$ and $p_{0t}(\boldsymbol{x}_t|\boldsymbol{x}_0)$ is the transition density of $\boldsymbol{x}_t$ from the initial timestep 0 to $t$. This object can be interpreted as a noise-matching loss in DDPM (Ho et al., 2020), which predicts noise components in $\boldsymbol{x}_t$ and can be illustrated as $\int_0^T \mathbb{E}_{\boldsymbol{x}_0 \sim p_0, \epsilon \sim \mathcal{N}(0,\mathbf{I})}[||\epsilon_\theta(\sqrt{\bar{\alpha}_t}\boldsymbol{x}_0 + \sqrt{1 - \bar{\alpha}_t}\epsilon, t) - \epsilon||_2^2]\mathrm{d}t$. This is regularly denoted as $\epsilon$-prediction parameterization (Ho et al., 2020; Jabri et al., 2022), and several other parameterizations including $F$-prediction (Karras et al., 2022; Kingma & Gao, 2023), score-prediction (Song et al., 2021) and velocity-prediction (Ma et al., 2024) have been proposed.

## 4 OBSERVATIONS

In this section, we examine the difficulties associated with learning denoising tasks across different timesteps, addressing inconsistencies in prior works regarding these difficulties. Our analysis is structured around two key aspects: 1) the convergence of loss and denoising performance across timesteps, providing insights into learning dynamics at various timestep stages in Section 4.1; and 2) the relative entropy change from $p_t$ to $p_{t-1}$ as a function of $t$, offering a quantitative measure of task difficulty progression over $t$ in Section 4.2. Upon integrating our findings, we establish a key conclusion: the learning difficulty for denoising tasks escalates as the timestep $t$ decreases.

### 4.1 ANALYSIS ON THE TASK DIFFICULTY IN TERMS OF CONVERGENCE SPEED

In this study, we analyze the convergence speed of loss and denoising performance across timesteps. To comprehensively cover various diffusion parameterizations, we utilized the notable frameworks DiT (Jabri et al., 2022) for $\epsilon$-prediction, EDM (Karras et al., 2022) for $F$-prediction, SiT (Ma et al., 2024) for velocity prediction. Detailed descriptions of the experimental setups are provided in Appendix B.

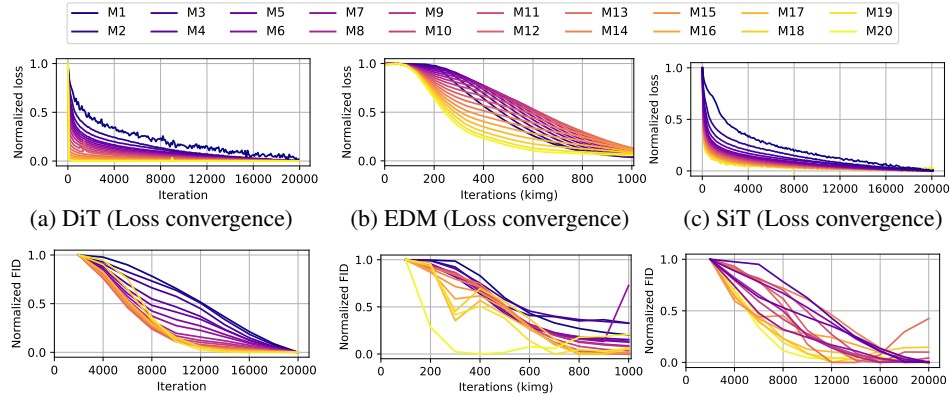

Figure 1: Loss and FID convergence plotted during training for each diffusion model $M_i$ in DiT, EDM, and SiT. Since the loss scale for each model is different, we show the normalized value. We observe that as $i$ increases (i.e., corresponding to larger denoising timesteps), the loss converges more rapidly, and this convergence speed correlates with that of the FID scores.

**Convergence speed on loss.** First, we analyze convergence characteristics of training loss across timesteps $t$. We divided whole timesteps $[0, T]$ into 20 uniformly divided intervals and trained 20 models $\{M_i\}_{i=1}^{20}$ where $i$-th model learns denoising tasks in $[\frac{i-1}{20}T, \frac{i}{20}T]$ for DiT and SiT, $[\Phi^{-1}(\frac{i-1}{N}), \Phi^{-1}(\frac{i}{N})]$ for EDM where $\Phi^{-1}$ is the inverse cumulative distribution function of the Gaussian distribution. During training, we tracked the loss values through iterations and plotted their convergence speed by normalizing their value in Figs. 1a-1c. As shown in the results, it is apparent that as $i$ increases towards $i = 20$, the convergence accelerates in both DiT, EDM, and SiT, suggesting that models learning larger timesteps can reach convergence more swiftly and reinforcing the notion that denoising tasks with larger timesteps are less difficult.

**Convergence speed on denoising performance.** We also delve deeper into a convergence of denoising performance according to timesteps with 20 distinct models $\{M_i\}_{i=1}^{20}$. To evaluate the performance of denoising tasks of each model, we generated samples where $M_i$ was employed for denoising within the timesteps that it was trained on, while a diffusion model learned whole timesteps handled denoising for the remaining timesteps as in (Go et al., 2023a). Then, the performance of the denoising capability of $M_i$ can be quantitatively measured using the FID score (Heusel et al., 2017), enabling us to observe the performance convergence of each model on denoising tasks throughout the training process. Figures 1d-1f depict this convergence. They illustrate that, similar to loss convergence experiments, denoising performance converges more swiftly for models $M_i$ with larger $i$ values, as observed across the DiT, EDM, and SiT. These results also suggest that models trained on later timesteps, indicated by larger $i$ values, achieve faster convergence, highlighting easier task difficulty at larger timesteps.

## 4.2 EXPLORATION ON DIFFICULTIES OF DENOISING TASKS

Beyond empirical convergence metrics, we also delve into analyzing the relative entropy between $p_t$ and $p_{t-1}$ to better understand task difficulties from a distributional perspective. The training of diffusion models implicitly involves learning the distribution of the reverse process of the corresponding SDE. To be specific, the transition probability of the reverse process is expressed as a conditional normal distribution whose mean parameter is modeled by neural networks, and they are thereby trained to learn the dynamics of the reverse process (Ho et al., 2020). Furthermore, an unconditional distribution of $x_t$ can be obtained by marginalizing transition densities over the prior distribution, indicating that information on the dynamics of the marginal distribution is fed to neural networks (Song et al., 2021).

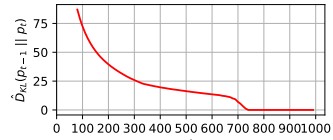

Figure 2: The KLD of $p_{t-1}$ from $p_t$ against denoising timestep. As the timestep increases, the dynamics decrease.

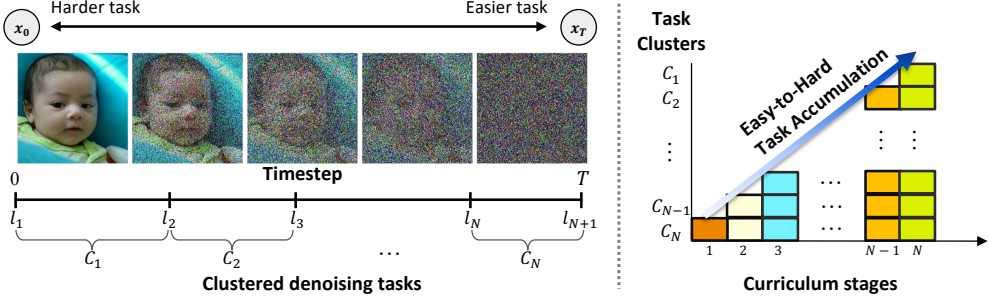

Figure 3: The overview of our curriculum learning approach for diffusion models. **(Left)** We divide the timesteps into $N$ clusters, $C_1, ..., C_N$, with the difficulty of denoising tasks increasing from $C_N$ (easiest) to $C_1$ (hardest). **(Right)** As the curriculum progresses, learning accumulates harder task clusters, gradually increasing task difficulties.

To analyze the relationship between the dynamics of the unconditional distribution and the rate of loss convergence, we use the Kullback-Leibler (KL) divergence of $p_{t-1}$ to $p_t$, $D_{KL}(p_{t-1}||p_t)$, as a quantitative measure. It is a pertinent divergence in that the training mechanism of diffusion models involves maximizing the likelihood of the reverse process. The KL divergence $D_{KL}(p_{t-1}||p_t)$ is given by $D_{KL}(p_{t-1}||p_t) = \mathbb{E}_{\boldsymbol{x} \sim p_{t-1}}\left[\log\left(\frac{p_{t-1}(\boldsymbol{x})}{p_t(\boldsymbol{x})}\right)\right]$. Moreover, the distribution $p_t$ of $\boldsymbol{x}$ at $t$ is expressed by $p_t(\boldsymbol{x}_t) = \int p_{0t}(\boldsymbol{x}_t|\boldsymbol{x}_0 = \boldsymbol{y})p_0(\boldsymbol{y})\mathrm{d}y = \mathbb{E}_{\boldsymbol{x}_0 \sim p_0}[p_{0t}(\boldsymbol{x}_t|\boldsymbol{x}_0)]$. However, since the explicit density form of $p_0$ is unknown and it is computationally infeasible to estimate high-dimensional integrals, we approximate them through unbiased estimators (details in Appendix C). The empirical results of $D_{KL}(p_{t-1}||p_t)$ for $64 \times 64$ image data are given in Fig. 2. As seen, the relative entropy tends to decrease as $t$ increases (i.e., $D_{KL}(p_{s-1}||p_s) \leq D_{KL}(p_{t-1}||p_t)$ for $s \leq t$), which is consistent with the results in Fig. 1.

This observation may stem from the inherent low-dimensional manifold of image data. As is well-known (e.g., (Ruderman, 1994)), the image data is distributed on a relatively low-dimensional manifold with a narrow support and a highly peaked multi-modal structure. On the other hand, as Gaussian noise is iteratively added, the distribution of $\boldsymbol{x}_t$ approaches the independent Gaussian distribution in the ambient space. Consequently, the support of the manifold broadens and the score function becomes regular over the ambient space with increasing $t$. This nature of the unconditional distribution may cause the relative entropy from $p_t$ to $p_{t-1}$ to decrease with $t$, indicating that it is more difficult to accurately represent the dynamics of the reverse process at small $t$. More discussion is in Appendix C.

## 5 METHODOLOGY

In Section 4, we observe that denoising tasks at smaller $t$ are more difficult to learn by models. From these order of difficulties in denoising tasks, we propose the incorporation strategy of an easy-to-hard training scheme, that has demonstrated its effectiveness in curriculum literature (Bengio et al., 2009; Hacohen & Weinshall, 2019; Kong et al., 2021), for improving diffusion models' training.

### 5.1 DESIGN OF CURRICULUM LEARNING IN DIFFUSION MODELS

As we observed in Section 4, difficulties in denoising tasks increase as $t$ gets smaller. To utilize an easy-to-hard curriculum learning approach, we first divide the entire range of timesteps into $N$ clusters, denoted as $\{C_i\}_{i=1}^N$, where each cluster $C_i$ spans an interval $[l_i, l_{i+1}]$, ensuring $l_i < l_{i+1}$, with $l_1 = 0$ and $l_{N+1} = T$, as shown on the left side of Fig. 3. The curriculum for training is constructed by regarding these task clusters as unit tasks, starting from the least challenging (the $N$-th cluster $C_N$) and advancing towards the most difficult (the first cluster $C_1$), through $N$ distinct stages. Specifically, in the $n$-th curriculum stage, we jointly train the model with denoising tasks sampled from the clusters $\bigcup_{j=N-(n-1)}^N C_j$ as illustrated in the right side of Fig. 3. The transition of the curriculum stages is determined by the pacing function, which will be discussed in the next section. After completing these $N$-stages of curriculum learning, the model continues to learn across the entire range of timesteps, $\bigcup_{j=1}^N C_j$, same as standard diffusion training.

The next consideration involves determining the boundaries for each cluster $l_i$. A straightforward approach is to uniformly divide the entire timestep interval $[0, T]$ as $C_i = [\frac{(i-1)\cdot T}{N}, \frac{i\cdot T}{N}]$ for $i = 1, 2, \ldots, N$. However, this method does not account for variations in noise levels across different timesteps. Therefore, to address this issue more effectively, we adopt an SNR-based interval clustering technique as used in (Go et al., 2023a), which aligns the clustering with the actual changes in noise levels, potentially enhancing curriculum learning adaptability to varying noise conditions.

For EDM (Karras et al., 2019) which operates based on the noise level $\sigma$ rather than the timestep $t$, and where $\sigma$ is sampled from a log-normal distribution such that $\log(\sigma) \sim \mathcal{N}(P_{\text{mean}}, P_{\text{std}}^2)$ during training, our clustering strategy for timesteps cannot be directly transposed. Given the log-normal distribution of $\sigma$, dividing it directly is impractical because $\sigma$ can extend over a wide range of values. Instead, we adapt our clustering approach to suit the log-normal characteristics by defining noise level clusters $C_i$. Specifically, we delineate $C_i = [\Phi^{-1}(\frac{i-1}{N}), \Phi^{-1}(\frac{i}{N})]$, where $\Phi^{-1}$ is the inverse cumulative distribution function (quantile function) of the Gaussian distribution $\mathcal{N}(P_{\text{mean}}, P_{\text{std}}^2)$. This method segments the noise levels into intervals by reflecting their probabilistic distribution.

## 5.2 Pacing Strategy of Curriculum

To effectively train the diffusion model according to the provided curriculum design, it is crucial to define a suitable *pacing function* for determining the transition of each $N$ distinct curriculum. Training for a fixed number of iterations for each curriculum stage is the simplest implementation (We also contain this method in experiments as *'NaiveCL'* in Section 6.2). However, the convergence rate of each curriculum phase varies significantly, as demonstrated in Fig. 1. Hence, we propose adopting an adaptive number of iterations for each curriculum, akin to the varied exponential pacing approach explored by Hacohen *et al.* (Hacohen & Weinshall, 2019). Our pacing function utilizes the training loss to determine transition moments and transitions to the next stage occur when the training loss converges at the current stage. Specifically, we introduce the maximum patience iteration $\tau$, and if the loss does not improve consecutively for $\tau$, the current curriculum stage is terminated, and the subsequent curriculum stage is initiated. Here, the maximum patience is a fixed hyper-parameter, and the detailed process and overall curriculum learning procedure are outlined in Algorithm **??** and **??** in Appendix D, respectively.

## 6 Experimental Results

In this section, we present experimental results to validate the effectiveness of our method. The advantages of our curriculum method, **1) Improved Performance**, **2) Faster Convergence**, and **3) Orthogonality with Existing Improvements**, are validated in this section. To begin, we outline our experimental setups in Section 6.1. Then, we provide the results of the comparative evaluation in Section 6.2, showing that our curriculum approach significantly improves the quality of generated samples compared to the baseline. Finally, analyses of our method are illustrated in Section 6.3 to deeply understand the effectiveness of our method.

## 6.1 Experimental Setup

Here, we provide experimental setups concisely. Detailed setups are presented in Appendix E.

**Evaluation protocols.**  For our comprehensive evaluation of various methods, we employed three distinct image-generation tasks: **1) Unconditional generation** with the FFHQ dataset (Karras et al., 2019), **2) Class-conditional generation** with CIFAR-10 (Krizhevsky et al., 2009) and ImageNet (Deng et al., 2009) datasets, and **3) Text-to-Image generation** with MS-COCO dataset (Lin et al., 2014). In 2) and 3) setups, we applied classifier-free guidance (Ho & Salimans, 2022).

**Target models.**  We employed three exemplary diffusion architectures for experiments: DiT (Peebles & Xie, 2022), which integrates latent diffusion models (Rombach et al., 2022) with Transformer architectures (Vaswani et al., 2017) parameterized as $\epsilon$-prediction, EDM (Karras et al., 2022), which focuses on pixel-level diffusion utilizing UNet-based architectures (Ronneberger et al., 2015) parameterized as $F$-prediction, and SiT (Ma et al., 2024) for score- and velocity-prediction. For the text-to-image generation, we incorporated a CLIP text encoder (Radford et al., 2021) as described in DTR (Park et al., 2024b).

Table 1: We evaluated unconditional image generation on FFHQ with DiT-B, EDM, and SiT-B, class-conditional image generation on ImageNet and CIFAR10 with DiT-L and EDM, respectively, and text-conditional image generation on MS-COCO with DiT-B. Note that our curriculum learning for diffusion models improves substantial performance without any additional parameters.

| $\epsilon$-prediction | | | | | | |
|---|---|---|---|---|---|---|
| Model | FFHQ 256×256 | ImageNet 256×256 | | | | COCO 256×256 |
| | FID↓ | FID↓ | IS↑ | Prec↑ | Rec↑ | FID↓ |
| DiT (Vanilla) | 10.49 | 11.18 | 146.95 | 0.75 | 0.47 | 7.62 |
| DiT + NaiveCL | 7.95 | 11.90 | 151.66 | 0.75 | 0.47 | 7.71 |
| **DiT + Ours** | **7.55** | **8.18** | **186.37** | **0.79** | 0.47 | **7.51** |

| $F$-prediction | | | Velocity-prediction | | Score-prediction | |
|---|---|---|---|---|---|---|
| Model | FFHQ 64×64 | CIFAR10 32×32 | Model | FFHQ 256×256 | Model | FFHQ 256×256 |
| | FID↓ | FID↓ | | FID↓ | | FID↓ |
| EDM (Vanilla) | 2.93 | 2.67 | SiT (Vanilla) | 7.44 | SiT (Vanilla) | 9.64 |
| EDM + NaiveCL | 3.13 | 2.88 | SiT + NaiveCL | 7.69 | SiT + NaiveCL | 9.77 |
| **EDM + Ours** | **2.71** | **2.44** | **SiT + Ours** | **6.95** | **SiT + Ours** | **9.15** |

## 6.2 COMPARATIVE RESULTS

In this section, we assess the effectiveness of our curriculum-based training approach. For a thorough comparison, we examine three distinct training variants, with further details provided in Appendix E: To achieve this, we compare three variants of training: *1) Vanilla*: This term refers to diffusion models trained using conventional methods without any curriculum learning strategies; *2) NaiveCL*: In this variant, we incorporate a basic curriculum learning strategy, which simply repeats the same number of iterations for each stage across an $N$-stage process and does not employ SNR-based clustering; *3) Ours*: This denotes our proposed curriculum approach, which is designed to enhance the training process of diffusion models by systematically structuring the learning stages.

**Quantitative evaluation.** We quantitatively validate the effectiveness of our methods with various architectures-DiT (Peebles & Xie, 2022), EDM (Karras et al., 2022), and SiT (Ma et al., 2024)- and tasks including unconditional, class-conditional, and text-to-image generation. Table 1 shows the results, confirming two empirical observations: 1) *NaiveCL* fails to consistently achieve improved performance compared to *Vanilla*, and 2) our approach outperforms both *NaiveCL* and *Vanilla*. Regarding the first observation, *NaiveCL* shows inconsistent improvements due to

Table 2: Evaluating the effectiveness of curriculum learning with extended training iterations on the ImageNet 256x256 dataset using the DiT-L architecture.

| FID↓   Iteration 
 Method | 400k | **2M** |
|---|---|---|
| DiT (Vanilla) | 11.18 | 7.84 |
| **DiT + Ours** | **8.18** | **6.24** |

its lack of robust adaptation on incorporating task difficulties in various task conditions. In contrast, our method demonstrates superior performance across all scenarios by improving the clustering and pacing of curriculums. Consequently, our approach consistently achieves significant performance enhancements across all metrics on four datasets: FFHQ (Karras et al., 2019), ImageNet (Deng et al., 2009), CIFAR-10 (Krizhevsky et al., 2009), and MS-COCO (Lin et al., 2014), illustrating its effectiveness regardless of data or model used.

Showing the results of longer training might demonstrate the robustness of our method in more extended training scenarios. We trained DiT-L/2 with 2M iterations and reported the results in Table 2. Our model consistently outperformed the baseline, demonstrating its effectiveness even in prolonged training. Therefore, our method proves to be robust and effective for longer training durations.

**Qualitative evaluation.** Due to space constraints, we illustrate a detailed collection of generated examples in Appendix F. In summary, our curriculum methodology demonstrates a notable enhancement in the quality of the images produced, when compared to *NaiveCL* and *Vanilla*.

## 6.3 ANALYSIS

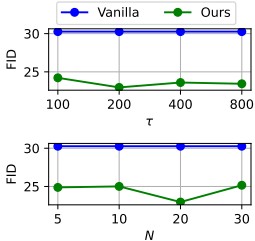

To elucidate our curriculum approach's effectiveness, we present a series of analytical studies. All the analysis is conducted by using the DiT-B model on the ImageNet dataset.

**Effects of $N$ and $\tau$.** We examined the robustness of the proposed curriculum training with respect to hyper-parameters: the number of clusters $N$ and the maximum patience $\tau$. As shown in Fig. 4, our method consistently outperforms the vanilla model, and the best result is observed at $N = 20, \tau = 200$. It shows that as $\tau$ increases, it may lead to overtraining due to excessive iterations for each task, whereas as $\tau$ decreases, curriculums may not be sufficiently trained. Furthermore, when the entire range of timesteps is finely partitioned (i.e., with an increase in $N$), each cluster becomes excessively granular, resulting in suboptimal performance. Conversely, with a decrease in $N$, tasks that should be in distinct clusters are learned together, forming a coarser cluster, which also leads to suboptimal outcomes. Overall, our method outperforms vanilla training across a range of hyperparameters, demonstrating the robustness of our approach.

Figure 4: Ablation study on $N$ and $\tau$. We use DiT-B on ImageNet 256×256.

**Effects of Curriculum Design.** In our curriculum design, we initially partitioned the entire set of timesteps into $N$ clusters using SNR-based clustering, organizing the curriculum from easy to hard clusters. To thoroughly assess the impact of each component, we conducted the ablation study as shown in Table 3. Firstly, we investigated the effect of curriculum learning via comparison with an anti-curriculum approach (Hacohen & Weinshall, 2019), which progresses from hard to easy clusters, unlike conventional curriculum learning. While both training methods in (b2) appear to enhance performance compared to vanilla training (a), anti-curriculum training cannot consistently guarantee performance improvement concerning the curriculum design as shown in (b1). In contrast, the proposed curriculum learning method (c1, c2) consistently exhibited performance improvement even with the uniformly partitioned clusters. Besides, with findings that utilizing SNR-clustering was more effective, clustering with the actual changes in noise levels enhanced the curriculum learning adaptability.

Table 3: Comparative results on various curriculum designs.

| Class-Conditional ImageNet 256×256. | | | | |
| --- | --- | --- | --- | --- |
| Curriculum Design | FID↓ | IS↑ | Prec↑ | Rec↑ |
| (a)  Vanilla | 30.27 | 60.06 | 0.55 | 0.52 |
| (b1)  + anti-curriculum + uniform | 31.12 | 62.80 | 0.55 | 0.53 |
| (b2)  + anti-curriculum + SNR | 27.74 | 68.10 | 0.58 | 0.52 |
| (c1)  + curriculum + uniform | 25.01 | 71.99 | 0.58 | 0.53 |
| (c2)  + curriculum + SNR | **22.96** | **75.98** | **0.62** | 0.52 |

**Visualization of curriculum.** To gain deeper insights into the functioning of our curriculum pacing, we plotted loss metrics against curriculum phases, as illustrated in Fig. 5. During the curriculum training, tasks progressively transition from the easiest to the most challenging, with varying amounts of iterations for each task based on the pacing function. The training loss decreased during each curriculum phase but increased after curriculum changes via the pacing function due to the inclusion of a newly added task in the updated curriculum. Additionally, as $\tau$ increases, the curriculum phases change more gradually, highlighting the role of $\tau$ in controlling the pace of curriculum transitions.

**Analysis on convergence speed.** As demonstrated in previous works (Bengio et al., 2009; Hacohen & Weinshall, 2019), the adoption of curriculum learning can lead to faster convergence in model performance. To illustrate the efficacy of our approach in this regard, we plotted the FID, IS, precision, and recall calculated over 10,000 samples across the training iterations, as depicted in Fig. 6. We observed the models trained through the proposed curriculum learning method converge faster than vanilla models, regardless of evaluation metrics. Notably, our approach achieves these improvements without requiring additional parameters or training iterations, thereby significantly saving time and computational resources.

**Effectiveness on various sizes of models** To verify the generalizability of our method across different model sizes, we evaluated the performance gains achieved using our curriculum learning approach on various scales of the DiT model: DiT-S (small), DiT-B (base), and DiT-L (large). Table. 4 shows that the proposed curriculum learning for diffusion model improves the performance regardless of the model size. Moreover, it is notable that larger models exhibit a more substantial performance enhancement: DiT-S improved by 8% in terms of FID, while DiT-B and DiT-L showed improvements

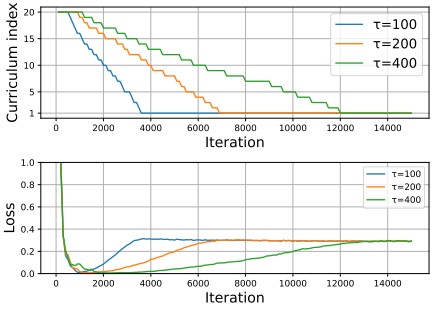

Figure 5: We visualized the curriculum transition and the corresponding loss across iterations ($N = 20$). To make the loss graph more easily readable, the y-axis was truncated to 1.0.

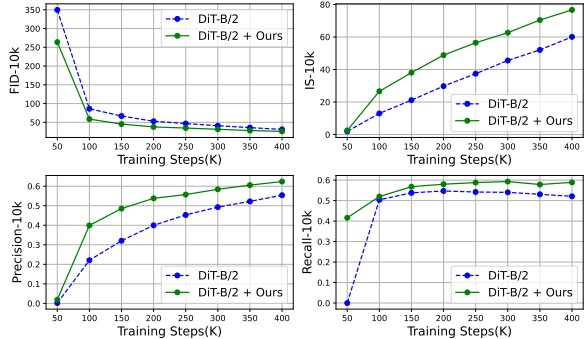

Figure 6: The models trained using the proposed curriculum learning approach demonstrate faster convergence compared to vanilla models, irrespective of evaluation metrics.

Table 4: Note that the curriculum learning achieves consistent improvements across the model sizes.

| Class-Conditional ImageNet 256×256. | | | | |
|---|---|---|---|---|
| Model | FID↓ | IS↑ | Prec↑ | Rec↑ |
| DiT-S/2 | 43.30 | 33.63 | 0.42 | 0.54 |
| DiT-S/2 + Ours | **39.66** | **36.57** | **0.44** | 0.54 |
| DiT-B/2 | 30.27 | 60.06 | 0.55 | 0.52 |
| DiT-B/2 + Ours | **22.96** | **75.98** | **0.62** | 0.52 |
| DiT-L/2 | 11.18 | 146.95 | 0.75 | 0.47 |
| DiT-L/2 + Ours | **8.18** | **186.37** | **0.79** | 0.47 |
| DiT-XL/2 | 9.40 | 166.83 | 0.77 | **0.49** |
| DiT-XL/2 + Ours | **7.57** | **234.93** | **0.82** | 0.48 |

Table 5: Note that the curriculum learning is compatible with the previous works such as the loss weighting (MinSNR) and architecture (DTR) study which, specified the multi-task learning for diffusion model.

| Class-Conditional ImageNet 256×256. | | | | | | | | |
|---|---|---|---|---|---|---|---|---|
| | DiT-B/2 | | | | DiT-B/2 + Ours | | | |
| | FID↓ | IS↑ | Prec↑ | Rec↑ | FID↓ | IS↑ | Prec↑ | Rec↑ |
| Vanilla | 30.27 | 60.06 | 0.55 | 0.52 | **22.96** | **75.98** | **0.62** | 0.52 |
| MinSNR (Hang et al., 2023) | 21.88 | 88.12 | 0.63 | 0.49 | **19.36** | **101.35** | **0.67** | 0.49 |
| DTR (Park et al., 2024b) | 15.77 | 89.89 | 0.68 | 0.52 | **15.33** | **91.39** | 0.68 | 0.52 |

of 24% and 27%, respectively. These findings validate the efficacy of our curriculum approach across a diverse range of model sizes, underscoring its generalizability to various model parameters.

**Orthogonality of Our Curriculum Approach** Lastly, we illustrate the seamless integration of our method with sophisticated training techniques such as DTR (Park et al., 2024b) and MinSNR (Hang et al., 2023). Initially, we observed that each sophisticated method yields a superior performance compared to the vanilla method. Meanwhile, as shown in Table. 5, the performance is significantly enhanced when we apply the proposed curriculum learning. Consequently, the curriculum approach proves to be compatible with previous promising methods such as loss weighting (MinSNR) and architectural enhancements (DTR), demonstrating our orthogonality with recent diffusion techniques.

**Additional experimental results** Due to limited space, we present additional experimental results in Appendix G. These results also support the effectiveness of our method, emphasizing the importance of curriculum approaches in diffusion training.

## 7 CONCLUSION

In this study, we tackle the challenge of denoising task difficulty within the diffusion model framework and introduce a novel task difficulty-based curriculum learning approach. To the best of our knowledge, we are the first to define task difficulty by considering both the convergence rates of loss and performance metrics. Moreover, in terms of data distribution analysis, we observe a reduction in relative entropy between consecutive probability distributions as timesteps progress. We believe that these observations might help reorganize the conflicts of previous works regarding denoising task difficulties. Building upon these insights, we propose a curriculum learning framework for diffusion models, comprising curriculum design and pacing strategies. Our experimental results convincingly demonstrate the efficacy of our approach across diverse diffusion model designs, datasets, and tasks. From these results, we emphasize that considering an order of learning denoising tasks is also a potential direction to improve training of diffusion models. In future research, for further enhancements, more advanced curriculum learning strategies such as self-pacing can be elaborated.

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
