# Denoising Task Difficulty-based Curriculum for Training Diffusion Models

**Jin-Young Kim**[*†]    **Hyojun Go**[*]    **Soonwoo Kwon**[*]    **Hyun-Gyoon kim**[1†]
Ajou University[1]
{seago0828, gohyojun15, swkwon.john}@gmail.com, hyungyoonkim@ajou.ac.kr

## A  Extended Related Work

### A.1  Analyzing Diffusion Model Behaviors in Each Timestep

In this section, we review works related to analyzing diffusion model behaviors in each timestep but not covered in detail in Section 2.1. Deja *et al.* (Deja et al., 2022) explore denoising during the backward diffusion process and observe that transition from denoising to generation exists in the backward process. Go *et al.* (Go et al., 2023) investigate the affinity between denoising tasks, showing that temporal proximal denoising tasks exhibit higher task affinity. Then, they also observe that simultaneously learning all denoising tasks by one model suffers from negative transfer. They can achieve better performance than standard diffusion training by alleviating negative transfer. Lee *et al.* (Lee et al., 2023) analyze frequency characteristics according to timesteps and observe that high-frequency components are lost as timesteps increase. From this observation, they propose a multi-architecture multi-experts diffusion model, which utilizes multiple denoiser models specialized in each timestep interval but utilizes a transformer-like model as the timestep increases. From observations that smaller and larger models produce similar latent noise, Pan *et al.* (Pan et al., 2024) propose T-Stitch, which leverages a pre-trained smaller model at the beginning of the backward process to accelerate the sampling speed. Xu *et al.* (Xu et al., 2023) investigate the average trace-of-covariance of training targets according to timesteps, showing that it peaks in the intermediate timesteps, causing unstable training targets. For more stable training targets, they utilize weighted conditional scores with a reference batch.

### A.2  Easy-to-hard training Strategy

Progressive distillation (Salimans & Ho, 2022) focuses on reducing the number of sampling steps by training the model to progressively skip more steps, while cascaded diffusion (Ho et al., 2022) aims to improve sample quality by progressively increasing the image resolution during training. Both methods concentrate on altering the model's behavior or structure to tackle specific challenges, such as efficiency or resolution enhancement. In contrast, our work identifies trends in task difficulty across timestep-wise denoising tasks and leverages these findings to propose an easy-to-hard training scheme. This training strategy directly addresses the order and structure of the learning process, optimizing task sequencing to enhance performance. This distinction emphasizes that our approach is fundamentally different from these methods, as it addresses a unique aspect of diffusion model training.

## B  Detailed Experimental Setups for Observation

In Section 4, we examined the difficulty of denoising tasks in terms of convergence with various models $\{M\}_{i=1}^{20}$, which are trained within specific timesteps $[\frac{i-1}{20}T, \frac{i}{20}T]$ for DiT and SiT, and $[\Phi^{-1}(\frac{i-1}{N}), \Phi^{-1}(\frac{i}{N})]$ for EDM where $\Phi^{-1}$ is the inverse cumulative distribution function of the Gaussian distribution. For the DiT architecture, we employed the DiT-B/2, whereas for EDM, we used the DDPM++ architecture. Both DiT and EDM models were trained on the FFHQ dataset, with a batch size of 256, for approximately 20,000 iterations and 4,000 kimg iterations (equivalent to processing 1 million images), respectively. This training was conducted until both loss and performance converged. As illustrated in Fig. A, we additionally plotted the iterations of each timestep interval when their loss values start to oscillate. We measured this by counting the number of times the loss value increased

---

[*]Co-first author    [†]Corresponding author

after the step reached 100. As shown in the results, losses of all timestep intervals are stabilized within 20K iterations, while the lower timesteps reach this point more slowly. This also suggests that the convergence speed of lower timesteps tends to exhibit a slower regime. To examine specifically at the observation of convergence, we also analyzed the convergence speed on the ImageNet dataset. As shown in Fig. B, we obtained similar results as on the FFHQ dataset. Configuration of training optimizers and learning rates are the same as setups in Section 6.

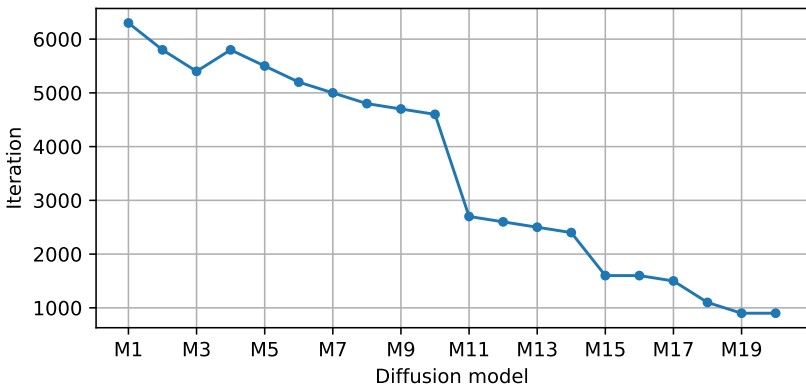

**Figure A:** Converged points are plotted during training for each diffusion model $M_i$ in SiT.

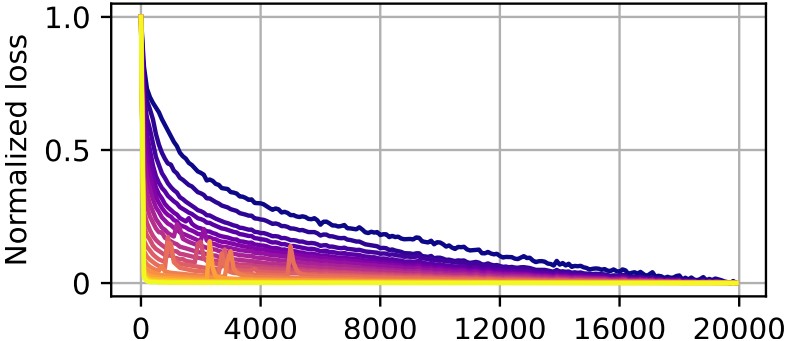

**Figure B:** Loss convergence plotted during training for each diffusion model $M_i$ in DiT on ImageNet dataset.

To evaluate the performance of diffusion models through the FID score of the generated images, performing a recursive denoising task from $T$ to zero is necessary, complicating the assessment using only $M_i$. Following (Go et al., 2023), we generated samples where $M_i$ was specifically utilized for denoising within its trained intervals. At the same time, the diffusion model is responsible for the denoising tasks across the entire range of timesteps. For this evaluation, we sampled 10K images using a DDPM sampler over 250 steps for DiT and SiT, and an Euler solver over 40 steps for the other models.

## C    APPROXIMATION OF KL DIVERGENCE OF $p_{t-1}$ AND $p_t$.

Here, we supplement the approximation of KL Divergence of $p_{t-1}$ and $p_t$ omitted in Section 4.2. To explore the difficulties of denoising tasks from the distributional viewpoint, we analyze the KL divergence of $p_{t-1}$ and $p_t$, $D_{KL}(p_{t-1}||p_t)$. However, due to the unknown explicit density form of

$p_0$, it is approximated through unbiased estimators as follows:

$$\hat{D}_{KL}\left(p_{t-1}\|p_t\right) = \frac{1}{M} \sum_{\substack{i \in \{1,2,\cdots,M\} \\ \boldsymbol{x}_i \sim p_{t-1}}} \log\left(\frac{p_{t-1}(\boldsymbol{x}_i)}{p_t(\boldsymbol{x}_i)}\right), \tag{1}$$

$$\hat{p}_t(\boldsymbol{x}_t) = \frac{1}{L} \sum_{\substack{j \in \{1,2,\cdots,L\} \\ \boldsymbol{y}_j \sim p_0}} p_{0t}(\boldsymbol{x}_t|\boldsymbol{x}_0 = \boldsymbol{y}_j), \tag{2}$$

where $\hat{D}_{KL}$ and $\hat{p}_t$ are unbiased estimators of $D_{KL}$ and $p_t$, respectively, and we choose Monte-Carlo estimators for them (Glasserman, 2004; McLeish, 2011).

We sampled 5,000 images to approximate the KL divergence, which is enough for Monte-Carlo sampling and might be no changes for larger samples. Despite the large amount of samples, the exploding appearance observed in Fig. 2 when $t$ is close to zero is due to the characteristics of the data distribution. The image data distribution has narrow support (roughly speaking, it is non-zero only within a narrow range) (Ruderman & Bialek, 1993; Karras et al., 2024). As $t$ increases, information about the original data distribution gradually diminishes with the modes in the distribution of $x_t$ vanishing towards zero.

Given this, when $t$ is close to zero (i.e. when the distribution of $x_t$ is still analogous to the original data distribution), the narrow support and the tendency to move towards zero give rise to a region where $p_{t-1}$ does not overlap with $p_t$. Consequently, when calculating the KL divergence $D_{KL}(p_{t-1}\|p_t) = E_{x \sim p_{t-1}}[\log(\frac{p_{t-1}(x)}{p_t(x)})]$, $x_{t-1}$ potentially falls outside the support of $p_t$, which leads to $p_t(x_{t-1}) = 0$ and numerical unstability. On the other hand, as $t$ increases, the accumulated noise broadens the support of $x$'s distribution, reducing the occurrence of zero values and stabilizing the numerical estimation.

## D  ALGORITHM

Due to the limited space of the main manuscript, we hereby present the step-by-step process of our method to supplement the details of our approach. The pacing function, which determines the moments to transit between curriculum stages is described in Algorithm 1. By incorporating this pacing function, the detailed procedure of our proposed curriculum learning method for training diffusion is illustrated in Algorithm 2.

---

**Algorithm 1** Pacing Function

**Input:** Current loss $L_{cur}$, Best loss $L_{best}$, Current patience $\tau_{cur}$, Maximum patience $\tau_{max}$, Current curriculum index $I_{cur}$

**Output:** Updated patience, Updated curriculum index

\# Reset patience
**if** $L_{cur} < L_{best}$ **then**
    **return** $0, I_{cur}$

**else**
    \# Proceed to next curriculum
    **if** $\tau_{cur} + 1 > \tau_{max}$ **then**
        **return** $0, I_{cur} - 1$
    \# Increase patience
    **else**
        **return** $p_{cur} + 1, I_{cur}$
    **end if**
**end if**

**Algorithm 2** Curriculum Learning

**Input:** Curriculum $\{C_i\}_{i=1}^N$, Pacing function $g$, Maximum patience $\tau_{max}$, Loss function $f$, Curriculum index $I_{cur} = N$, Best loss $L_{best} = \infty$, Model $M_\theta$
**while** $I_{cur} > 0$ **do**
    \# Mini-batch sampling
    $X \sim C_{I_{cur}}$
    \# Calculate Loss
    $L_{cur} = f(M_\theta(X))$
    \# Update model
    $\theta = \theta - \nabla_\theta L_{cur}$
    \# Pacing function
    $\tau_{cur}, I_{next} = g(L_{cur}, L_{best}, \tau_{cur}, \tau_{max}, I_{cur})$

    \# Update curriculum
    **if** $I_{cur} \neq I_{next}$ **then**
        $I_{cur} = I_{next}$
        $L_{best} = \infty$
    \# Update best loss
    **else if** $L_{cur} < L_{best}$ **then**
        $L_{best} = L_{cur}$
    **end if**
**end while**

---

## E  DETAILS ON EXPERIMENTAL SETUPS

**Evaluation metrics.**  To evaluate the performance of models, we utilized three metrics: FID (Heusel et al., 2017), IS (Salimans et al., 2016), and Precision/Recall (Kynkäänniemi et al., 2019). Specifically, we applied FID and IS to measure sample quality, while Precision is used to assess quality further

and Recall was utilized to evaluate the diversity of the generated samples in ImageNet setup. In other datasets, we employed FID to evaluate sample quality. Unless otherwise mentioned, we sampled 50K samples for evaluation. In tasks involving conditional generation, including class-conditional image generation (e.g. CIFAR-10, ImageNet) and text-to-image conversion (e.g. MS-COCO), we adapted the classifier-free guidance (Ho & Salimans, 2022) with a guidance scale of 1.5.

**Training details.** For training diffusion models, we utilized the AdamW optimizer (Loshchilov & Hutter, 2017) with a constant learning rate of 0.0001, and weight decay was not applied. The exponential moving average (EMA) to the model's weights was used to stabilize the training and the decay ratio was set to 0.9999. The batch size was set to 256, and we augmented the training data by a horizontal flip. While the diffusion timestep $T$ was configured as 1,000 for all experiments, we trained for 100K iterations for the FFHQ dataset (Karras et al., 2019), and 400K iterations for the ImageNet dataset (Deng et al., 2009) and MS-COCO dataset (Lin et al., 2014). The number of clusters $N$ was 20 unless otherwise specified. The maximum patience $\tau$ was varied across model sizes: it was set at 200 for DiT-S/2, DiT-B/2, and EDM, and 400 for DiT-L/2. EDM was trained using fp16, while the other models were trained using fp32. We used 8 A100 GPUs for all experiments.

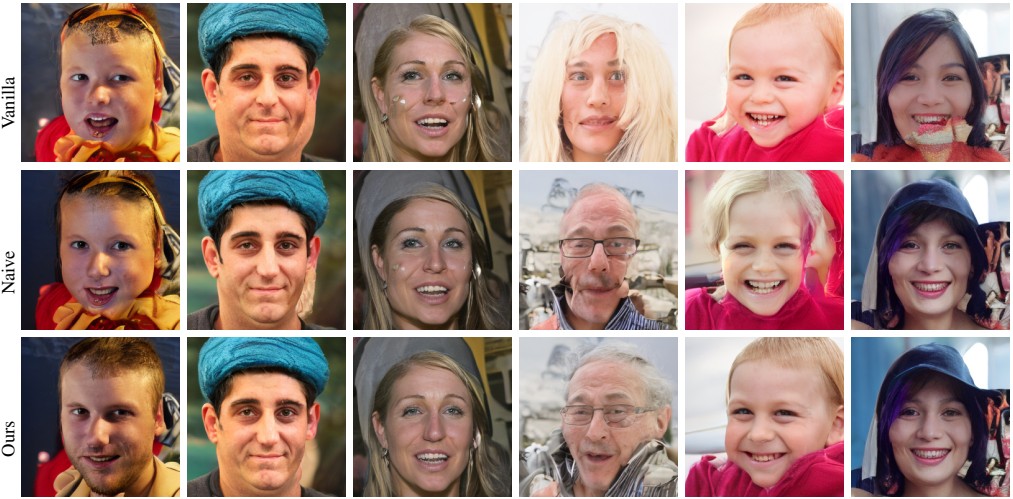

**Figure C:** Qualitative comparison between vanilla, naive curriculum, and ours on the FFHQ dataset.

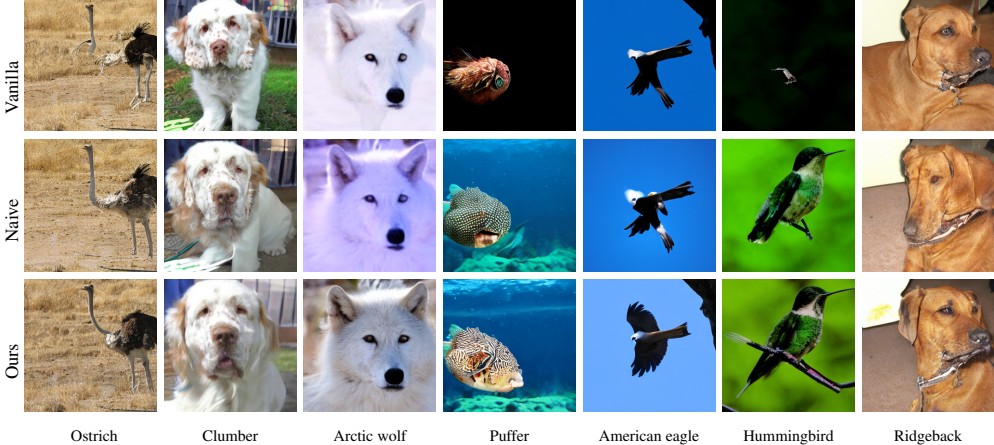

**Figure D:** Qualitative comparison between vanilla, naive curriculum, and ours on ImageNet dataset.

## F QUALITATIVE RESULTS

In this section, we present qualitative comparisons between three methods: 1) *Vanilla*, 2) *NaiveCL*, and 3) *Ours*, across the FFHQ, ImageNet, and MS-COCO datasets. All methods are evaluated using

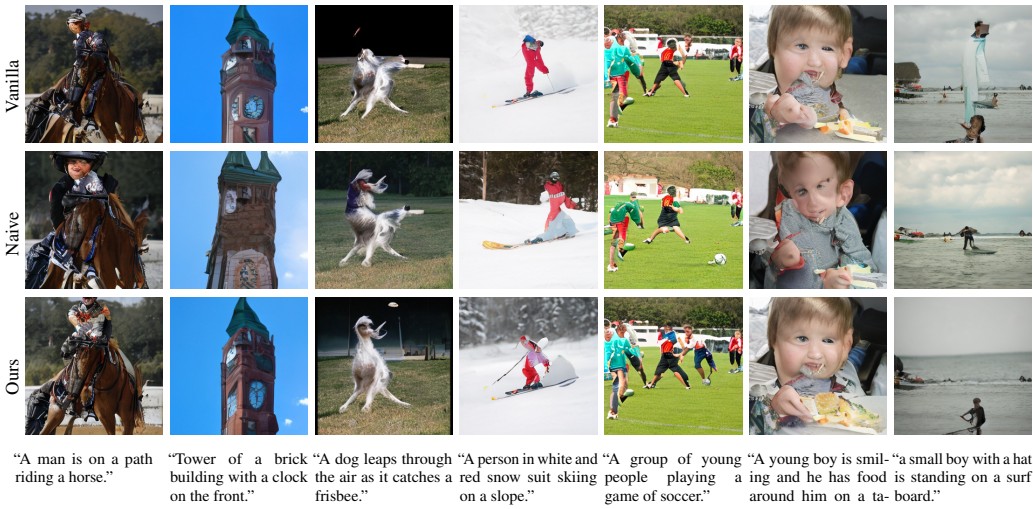

**Figure E:** Qualitative comparison between vanilla, naive curriculum, and ours on MS-COCO dataset.

DiT-B models, and the final trained models generate all samples shown in the results. As shown in the results in the following subsections, our approach can synthesize more accurate and realistic images compared to *Vanilla* and *NaiveCL*.

### F.1 QUALITATIVE EVALUATION ON THE FFHQ DATASET.

Figure C presents a qualitative analysis of the performance in unconditional facial image synthesis among the vanilla, the naive curriculum approach, and our method. Our approach demonstrates superior ability in generating realistic images.

### F.2 QUALITATIVE ANALYSIS ON THE IMAGENET DATASET.

In the conditional image synthesis, we exhibit the outcomes generated by the vanilla, the naive curriculum strategy, and our proposed method. Figure D clearly shows that our methodology surpasses the competing approaches in terms of quality.

### F.3 QUALITATIVE ASSESSMENT ON THE MS-COCO DATASET.

To further substantiate the effectiveness of our proposed technique, we conduct a qualitative comparison of the results in the text-to-image generation task among the vanilla, the naive curriculum method, and our own approach, as depicted in Fig. E.

## G FURTHER EXPERIMENTAL RESULTS

### G.1 CONVERGENCE SPEED ACROSS MODEL SIZE

By leveraging the advantages of curriculum learning in diffusion training, our method offers faster convergence than vanilla training. To further investigate this aspect, we measured FID-10K through training iterations for DiT-S and DiT-L. Figure F describes the results, showing that our curriculum approach achieves faster convergence in both models. These results also support the effectiveness of our method.

### G.2 ROBUSTNESS ON NOISE SCHEDULE

For a more comprehensive ablation study, we also trained the diffusion model with different noise schedules. In contrast to cosine scheduling, the $\beta_t$ is set by uniformly dividing the interval $[0.0001, 0.02]$, and the $C_i$ are obtained corresponding to SNR on a linear schedule. As shown in Table. A, our approach improves the performance with cosine and linear noise schedulers.

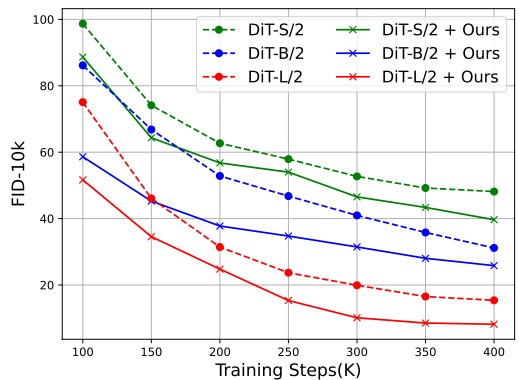

**Figure F:** We observed an increase in convergence speed across various model sizes when the proposed curriculum learning approach was applied.

**Table A:** Ablation study on noise scheduler. Note that our approach improves performance consistently across each scheduler.

| Class-Conditional ImageNet 256×256. | | | | | |
|---|---|---|---|---|---|
| Schedule | Method | FID↓ | IS↑ | Prec↑ | Rec↑ |
| cosine | Vanilla | 30.27 | 60.06 | 0.55 | **0.52** |
| | Ours | **22.22** | **75.98** | **0.62** | **0.52** |
| linear | Vanilla | 16.99 | 83.62 | 0.68 | **0.53** |
| | Ours | **16.03** | **87.66** | **0.69** | **0.53** |

### G.3 QUALITATIVE RESULTS FROM DIT-L/2 WITH 2M ITERATIONS

In Figures G-K, we present images generated by DiT-L using our curriculum training method for 2M iterations. The results demonstrate that our method produces highly realistic images.

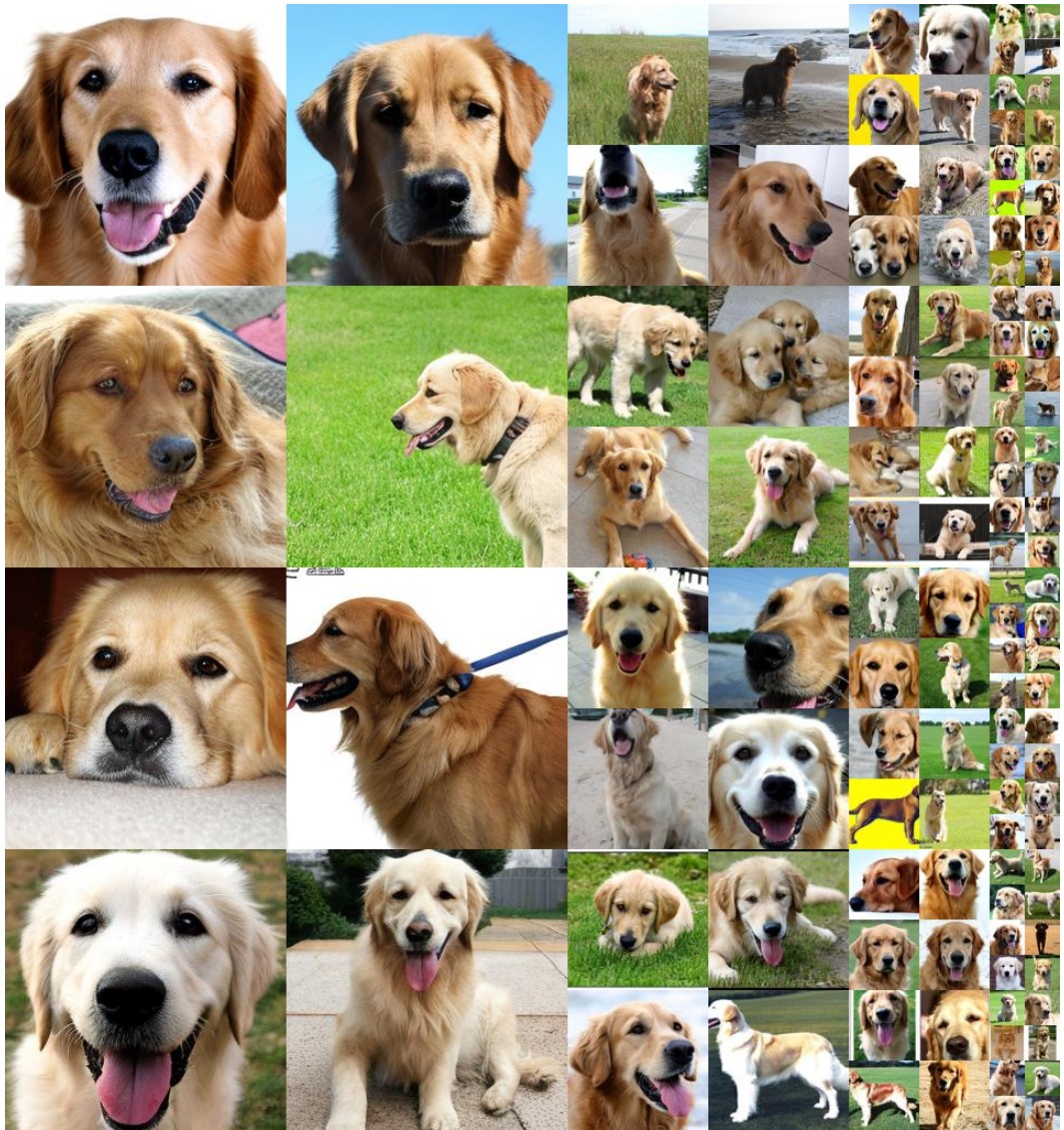

**Figure G:** Uncurated 256×256 DiT-L/2 samples.
Classifier-free guidanzce scale = 2.0.
Class label = "golden retriever" (207)

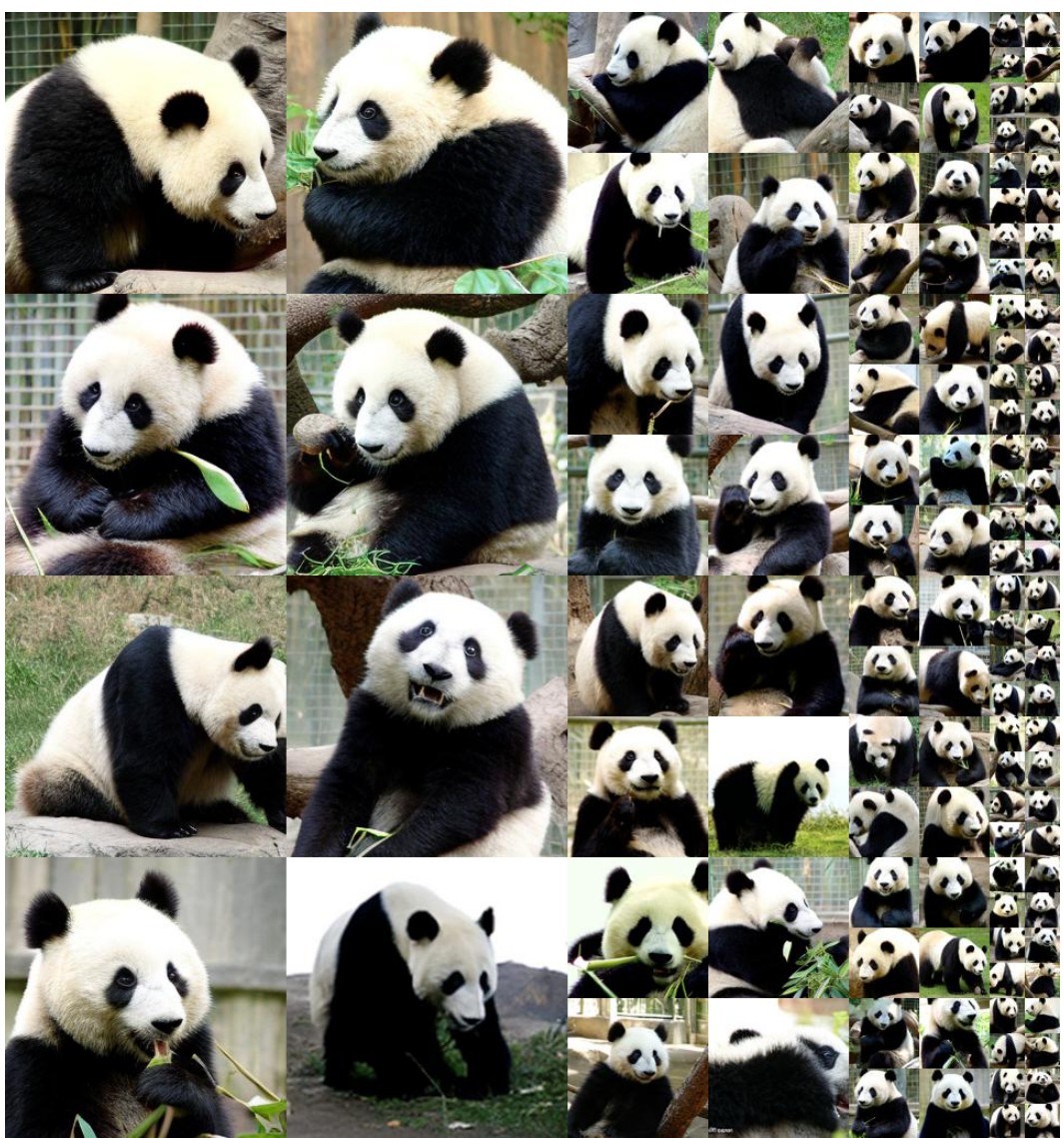

**Figure H:** Uncurated 256×256 DiT-L/2 samples.
Classifier-free guidance scale = 2.0.
Class label = "panda" (388)

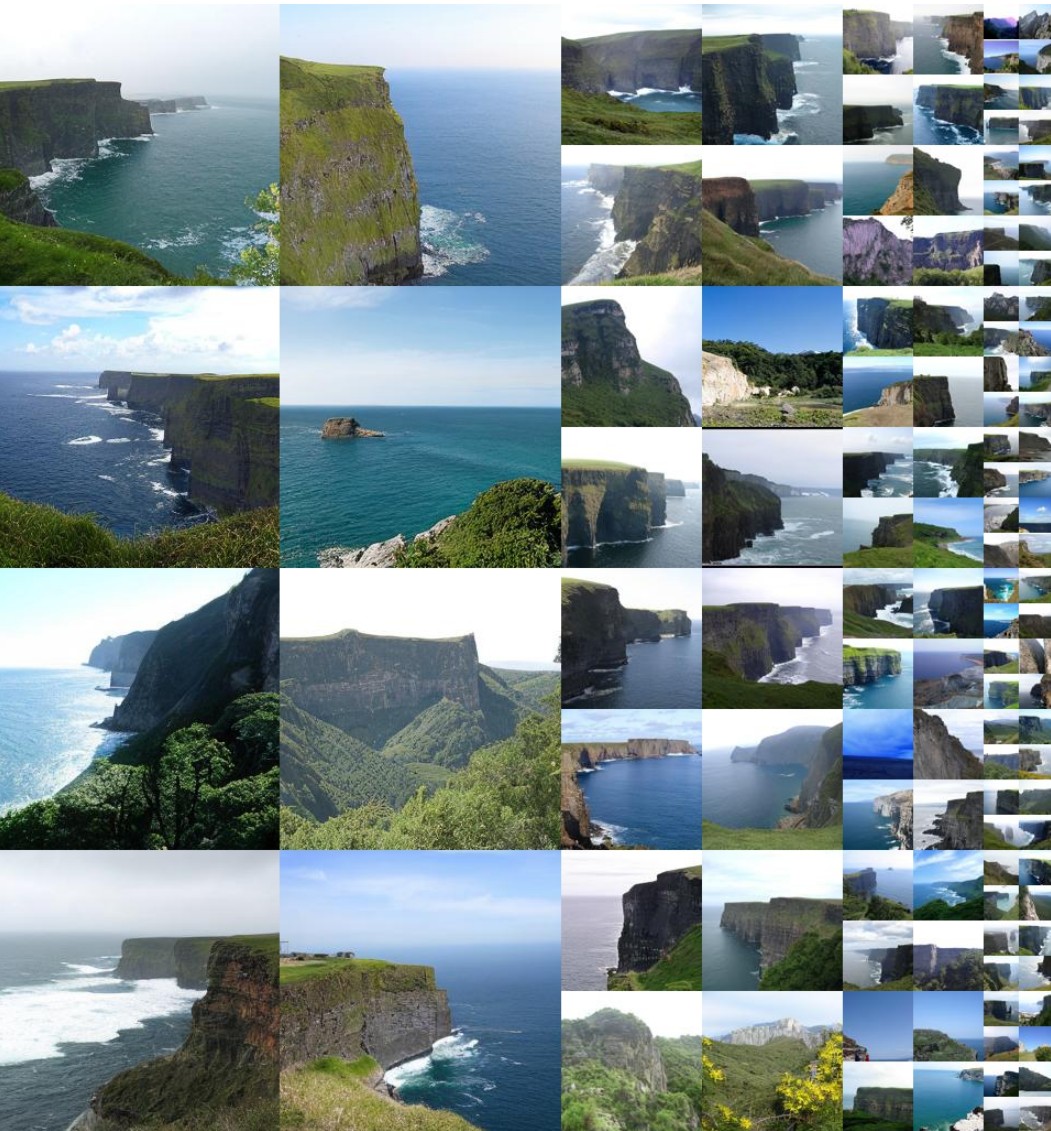

**Figure I:** Uncurated 256×256 DiT-L/2 samples.
Classifier-free guidance scale = 2.0.
Class label = "cliff drop-off" (972)

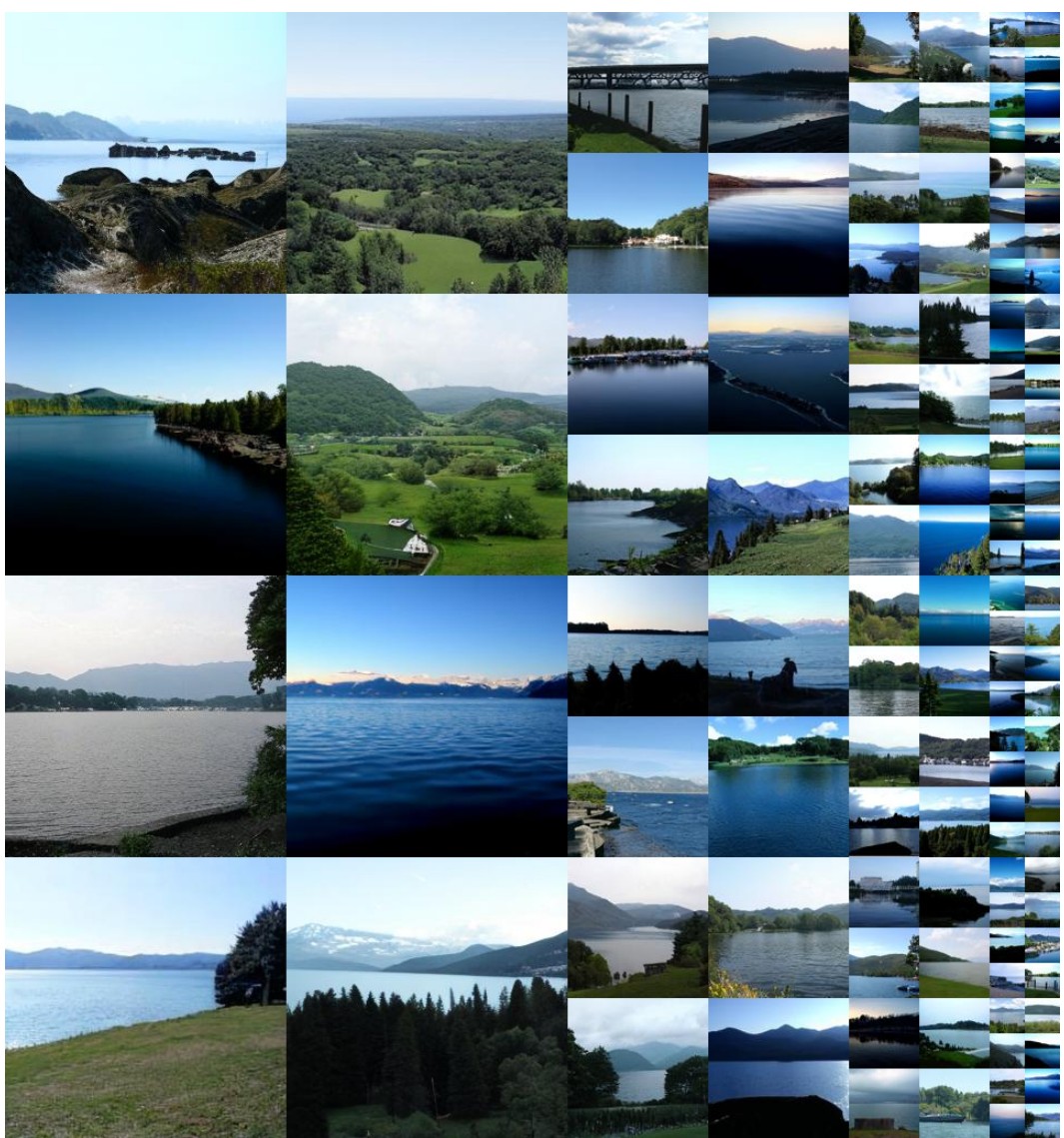

**Figure J:** Uncurated 256×256 DiT-L/2 samples.
Classifier-free guidance scale = 2.0.
Class label = "lake shore" (975)

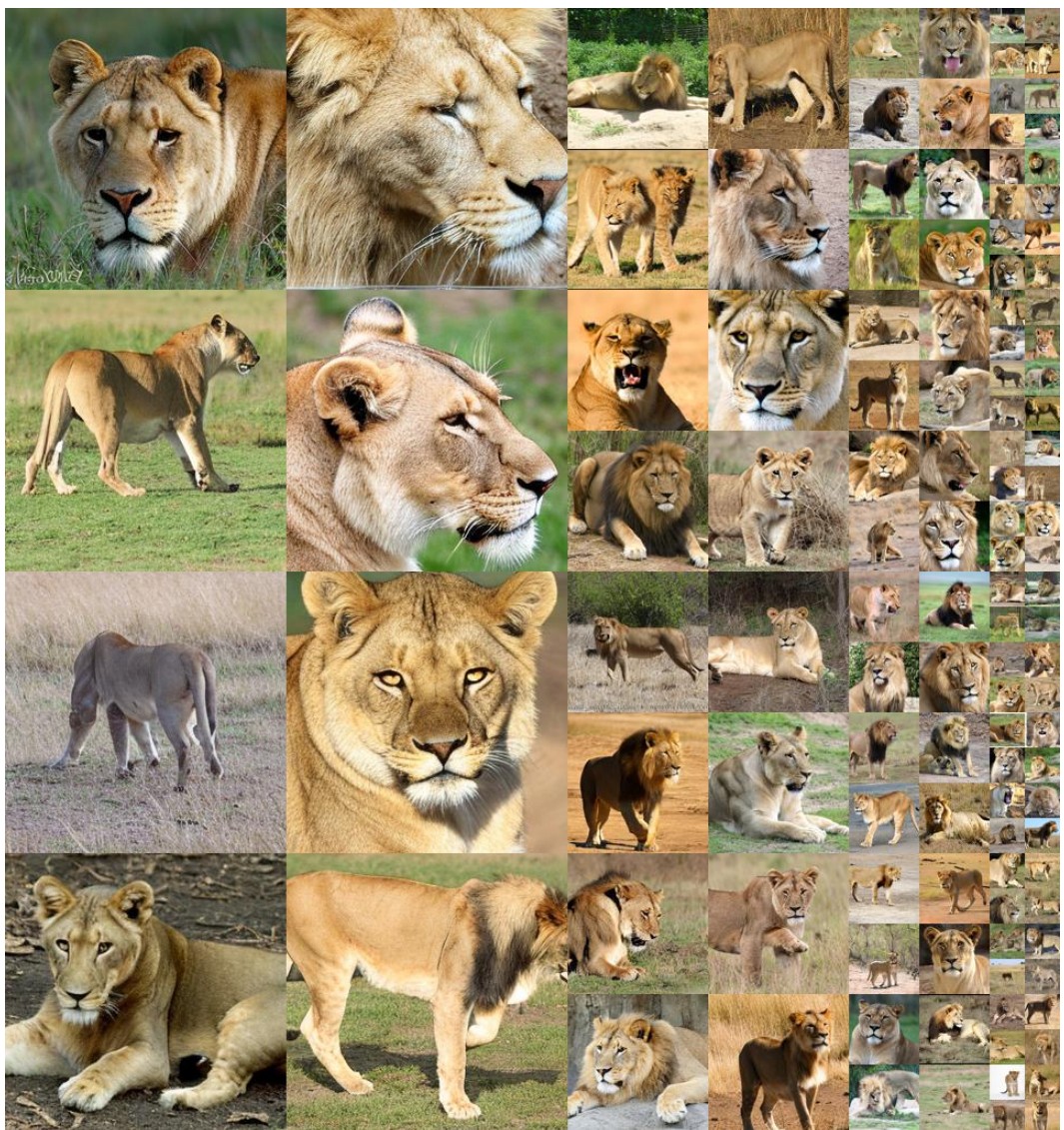

**Figure K:** Uncurated 256×256 DiT-L/2 samples.
Classifier-free guidance scale = 2.0.
Class label = "lion" (291)

## H    DISCUSSION ON SIMILARITY WITH THE PREVIOUS WORK

While both our work and (Go et al., 2023) explore the characteristics of denoising tasks in diffusion models, the aspects of exploration in each work are substantially different. The notion of task affinity introduced in (Fifty et al., 2021; Go et al., 2023) refers to how harmoniously the model can learn multiple tasks together. Specifically, their work focuses on identifying and mitigating conflicts between tasks, emphasizing task interactions and transferability by analyzing task similarities (e.g., gradient similarity or alignment). In contrast, our work explicitly quantifies the relative difficulty of individual denoising tasks across timesteps as a standalone property, independent of task interdependencies. The analysis of task difficulty in our work involves evaluating metrics such as loss behavior or convergence rates, directly reflecting the complexity of solving each task at different timesteps. Therefore, while (Go et al., 2023) addresses how tasks relate and interact during multi-task learning, our focus lies in systematically characterizing the intrinsic difficulty of tasks across timesteps in diffusion models.

## I    BROADER IMPACTS

Generative models, such as diffusion models, have the potential to significantly impact society, particularly through DeepFake applications and the use of biased datasets. One primary concern is the possibility for these models to amplify misinformation, which can erode trust in visual media. Additionally, if these models are trained on biased or deliberately altered content, they may unintentionally perpetuate and intensify existing social biases. This situation may result in the dissemination of incorrect information and the manipulation of public opinion.

## J    LIMITATIONS

In this work, we demonstrated the varying difficulties of denoising tasks through empirical results on various diffusion frameworks and proposed a curriculum learning approach that effectively enhances diffusion model training. While we have shown the robustness of our method's hyperparameters in improving vanilla diffusion training, there is potential for further improvement. Specifically, curriculum learning methods that utilize smaller hyperparameters and adjust dynamically based on the model itself could yield better results. We acknowledge the validity of this direction and consider it a promising avenue for future work.