# OpenReview forum: "Denoising Task Difficulty-based Curriculum for Training Diffusion Models"
_ICLR.cc/2025/Conference — ICLR 2025 Poster_

### Official Review · Reviewer_Sh9g · 2024-10-17

**Soundness:** 3
**Presentation:** 3
**Contribution:** 3
**Rating:** 6
**Confidence:** 4

**Summary:**

In this work, a new analysis of the levels of difficulty of different parts of diffusion model is presented followed by the introduction of new curriculum-based approach for improved training. Authors first show that it is harder to learn how to denoise samples that are less noisy in the diffusion process. On top of this observation they introduce a new method for the training of diffusion models, where model is first trained with only simpler timesteps, followed by the harder ones and full training. The extensive evaluation validates that such approach leads to better performance of the final model and faster convergence.

**Strengths:**

- The idea of curriculum learning for faster and more efficient diffusion training is interesting and, to the best of my knowledge, novel. The experimental evaluation is convincing as the method seems to improve the performance of the models across several benchmarks and with different architectures
- Submission provides interesting experiments in the preliminary observation. I particularly appreciate experiments where part of the sampling trajectory was generated using a separate model trained for a limited period of time. This experiment brings valid observations regarding how challenging the training of earlier/later diffusion steps is. However, there is still one question remaining - what is the influence of the training on the remaining steps on other steps - see the questions section.
- On top of the proposed method and main benchmarking, the authors present an extensive additional experiments section that provides an in-depth analysis of the solution’s strengths.

**Weaknesses:**

- I’m left with a single concern. Is it really thanks to the curriculum learning, or is it just important to first learn how to do denoising in the initial steps of the diffusion process - which define the mapping between random Gaussian noise and training data so that later training is easier? Driven by the confusing results of the evaluation presented in Figure 4, I lack one last experiment where the model is first trained using only timesteps from the C_N cluster followed by random ordering or standard training. Would it be significantly worse than the presented approach?
- In Section 5.2 there is a statement that “the convergence rate of each curriculum phase varies significantly, as demonstrated in Fig. 1.” - - - This is true, but I hope that I understand correctly that in the CL scenario, the model is first trained with the easy task, but then the same model is further finetuned with harder tasks. This should affect the convergence speed.
- Figure 4 is very puzzling. It suggests that it actually doesn’t matter how much we split the process used by the curriculum training, the results are almost identical except for the magical 20 splits used throughout the rest of the submission.
- Small errors/suggestions: Section 5 describes the Method rather than the Methodology. Table 2, Figure 4 are not within the specified margins

**Questions:**

- How splitting the process into separate models for separate parts affected the loss convergence? I can imagine that when training a single model on all of the steps, there is some positive/negative transfer between different tasks.
- Are all of the models in the evaluation section trained for the same number of steps? How is it achieved in the CL training with maximum patience iteration that introduces various number of training steps in each CL task?
- The results presented in Table 1 for some methods are relatively close to each other. How many examples were used to calculate FID, are those results statistically significant (as there are no confidence intervals)
- Were all of the models trained with a linear noise scheduler? How would the results be affected by changing it?

---

> ### Author Response · Authors · 2024-11-21
>
> We deeply appreciate insightful comments which are very helpful in making our work complete. We will address all raised concerns.
>
> --------
>
> ## **W1: Ablation on curriculum scheduling**
>
> Thank you for your insightful points. To address your concern, we conducted an additional experiment where the model was first trained using only the easiest cluster C_N  and then continued with a random ordering of timesteps on FFHQ with DiT-B. The results show the following FID scores: curriculum (FID: 7.55) > vanilla (FID: 10.49) > random (FID: 11.88) > anti-curriculum(FID: 15.53). These results demonstrate that curriculum learning consistently outperforms both random ordering and standard training, highlighting the effectiveness of the proposed approach beyond simply learning the easiest cluster first.
>
> --------
>
> ## **W2: Clarification of the relation between curriculum learning phases and convergence speed.**
> We want to clarify that the model is initially trained in a curriculum phase, starting with the easiest tasks and gradually incorporating harder tasks. After the curriculum phase, standard simultaneous training of all timesteps is conducted. In general (or in most cases), the gradual progression of curriculum learning completes far earlier than the final training iteration, as shown in Fig. 5, the curriculum phase completes before 14K steps, ensuring that standard training is included in the overall training process.
>
> --------
>
> ## **W3: Effect of the number of clusters**
> Thank you for your observation. The results may appear similar due to our reporting of only FID, but if you look at the additional metrics such as IS, precision, and recall in the table below, you will see that they reveal different patterns across the number of clusters N.
>
> |Class-Conditional ImageNet 256x256.|       |      |      |      |
> |------|---|---|---|---|
> |*N*|*FID*|*IS*|*Prec*|*Rec*|
> |5|24.88|68.68|0.58|0.51||
> |10|25.01|68.83|0.58|0.52|
> |20|22.96|75.98|0.62|0.52|
> |30|25.16|73.58|0.61|0.51|
>
> --------
>
> ## **W4: Small errors/suggestions**
> We appreciate your attention to detail and will ensure your suggestions are reflected in the revised version.
>
> --------
>
> ## **Q1: Negative transfer between different tasks**
> We propose a method where a single model is trained progressively, starting with easy denoising tasks and gradually incorporating harder ones, while also utilizing the same single model during sampling. This approach cannot be directly applied to training dedicated models for each task (such as Lee et al., 2024). Moreover, our method offers advantages in terms of memory and cost efficiency compared to approaches that rely on separate expert models for different tasks.
>
> --------
>
> ## **Q2: Detail of training**
>
> Thank you for your question. If not explicitly stated, all models were trained for 400K steps. As shown in Fig. 5, the curriculum phase completes before 14K steps, meaning that the total training iterations do not exceed this limit. When adjusting for patience, setting the total training iterations divided by the number of clusters ensures that the total steps are reached before the curriculum phase ends. This approach prevents exceeding the total training iterations before the curriculum training is completed.
>
> --------
> ## **Q3: The number of samples for evaluation**
>
> We used 50K samples to calculate FID, as described in Appendix E. Due to computational constraints, we could not perform repeated experiments to measure statistical significance. However, it is common in diffusion research not to conduct repeated experiments due to the high computational cost. Furthermore, we believe the effectiveness of our method is well-demonstrated through extensive experiments across various datasets, models, and model sizes.
>
> --------
>
> ## **Q4: Ablation study on noise scheduling**
>
> Thank you for your question. Ablation study results on noise scheduling can be found in Appendix G.2. Our proposed method demonstrates consistent improvement across both cosine and linear noise scheduling, indicating its robustness to different noise schedules.
>
> --------
>
> ## **References**
> Lee et al., Multi-Architecture Multi-Expert Diffusion Models, 2024

---

> > ### Comment · Reviewer_Sh9g · 2024-11-24
> > **Response to the authors**
> >
> > I am thankful for an additional experiment that clarifies my concern. I just want to make sure that I understand the notation correctly. In the setup called “random,” the model was first trained with the cluster C_N for the same amount of training steps as it would be for this part of the CL scenario. Then, it was further fine-tuned in the same way as the “vanilla” model for the remaining number of training steps. So, is it only the initial training with the easiest time steps breaking the model that the final FID is higher than for the vanilla model?
> >
> > W3: This is still puzzling for me. Do you have any hypothesis as to why there is higher precision of the generations when training the models with 20 clusters split, while it drops when splitting the process into 30?
> >
> > Q1: I think there is a misunderstanding, sorry for not being precise. In this question, I refer only to the analysis presented in Fig 1. As far as I understand, in this analysis, a set of 20 separate models was trained (line 236). My question is, what would be the loss convergence or task convergence if taking a model trained on all timesteps in a vanilla way and using it in the evaluation, for example, in the same way as M1? Would it be located below the current M1 curve? In other words, can we observe positive effects on the performance of the model applied in early timesteps because it was also trained to denoise at later steps? This question is purely curiosity-driven; it does not indicate any weakness in the submission.
> >
> >
> > Thank you for clarification regarding the remaining questions. I’m content with the response.

---

> > > ### Author Response · Authors · 2024-11-25
> > >
> > > We sincerely appreciate your prompt feedback and the engaging discussion. We would like to provide additional clarification regarding the reviewer’s questions:
> > >
> > > ---
> > >
> > > ### **Question: Notation Regarding "Random"**
> > > We sincerely apologize for any confusion in our previous response. The term "random" refers to a model trained on the task cluster $C_N$ first, followed by curriculum learning with randomly ordered remaining clusters, $C_1, \dots, C_{N-1}$. As highlighted in the results, random ordering in curriculum learning leads to performance degradation, underscoring the importance of our specific curriculum learning procedure in achieving improved results.
> > >
> > > Additionally, to address your concern, we conducted an experiment where the model was trained on $C_N$ initially, followed by standard training without a curriculum. In this setup, the FID score was **9.64**, reflecting a slight improvement over vanilla training. This result indicates that training on $C_N$ first is beneficial as it follows an "easy-to-hard" progression. However, the performance does not reach the level achieved by our curriculum learning method, demonstrating the necessity of our approach for higher performance.
> > >
> > > ---
> > >
> > > ### **W3: Performance Drop with 30 Clusters**
> > >
> > > The observed performance drop when increasing the number of clusters to 30 can be attributed to granularity. As the number of clusters increases, the tasks become overly fine-grained, leading the model to focus on specific subtasks within the cluster. This excessive granularity may prevent the clustering approach from achieving its full potential. To enhance the clustering approach, it appears that there is an optimal range of cluster granularity that balances task division effectively.
> > >
> > > ---
> > >
> > > ### **Q1: Suggestion Regarding Figure 1 Analysis**
> > > We are deeply thankful for this insightful suggestion. With your suggestion, we can more deeply understand the positive effects of each learning process.
> > > Unfortunately, due to the limited time available during the rebuttal period, we regret that we cannot incorporate the suggested analysis at this stage. However, we fully recognize its value and will make sure to include it in the final version of the paper.
> > >
> > > ---
> > >
> > > Thank you again for your valuable feedback. Please feel free to let us know if there are any additional questions or points requiring clarification.

---

### Official Review · Reviewer_xMw1 · 2024-10-28

**Soundness:** 3
**Presentation:** 3
**Contribution:** 2
**Rating:** 8
**Confidence:** 4

**Summary:**

The paper proposes a curriculum-based training schedule for diffusion models that trains the model on progressively increasing task difficulty, which corresponds to different bins of the diffusion noise schedule. The authors provide empirical evidence that task difficulty increases with decreasing time steps (as SNR increases), and provide a simple scheme of dividing the training into difficulty tasks. Results show that the scheme improves the convergence and quality of diffusion samples with no training overhead.

**Strengths:**

- Dividing training into difficulty phases determined by diffusion timestep clusters is novel to my knowledge.
- The paper is well structured and the method is well motivated. The presentation of evidence that difficulty varies depending on diffusion timestep (Sec 4) into the proposed method (Sec 5) is well thought and makes the paper flow nicely.
- Comprehensive experiments show improved performance over baselines at same number of training iterations, i.e., with no additional training overhead, with sensible ablations justifying design choices made by the authors.

**Weaknesses:**

- Overall the main novelty of the work is on the proposed training schedule, which is simple and on the lower side. Sec 4 is more of an empirical confirmation that lower noise levels is more difficult, which I believe is already well-known. I think this is not a big negative point however, as there is merit to simple ideas that work.
- Some missing citations as the general idea of training diffusion models from easy to difficult tasks is not new. The earliest and most influential ones to my knowledge are progressive distillation [1] (many to few sampling steps) and cascaded diffusion [2] (low to high-res).
- Sec 4.2, it is not clear to me why high KL between marginals of two time steps implies higher task difficulty. I assume it is because higher KL means the model has to make larger changes to the image between timesteps, thus making it more challenging. I think this can be more clearly stated.

[1] Salimans, Tim, and Jonathan Ho. "Progressive distillation for fast sampling of diffusion models." arXiv preprint arXiv:2202.00512 (2022).

[2] Ho, Jonathan, et al. "Cascaded diffusion models for high fidelity image generation." Journal of Machine Learning Research 23.47 (2022): 1-33.

**Questions:**

- Interesting that the anti-curriculum training (hard to easy) can also improve performance over vanilla, even if not consistently (Table 3). Do the authors have insight on why? This might come across as contradictory to the main claims of the paper and I suggest the authors explain it clearly to avoid confusing the reader.
- Might be small typo in line 294. As time step increases, the expression in parenthesis suggests KL increases which contradicts Fig 2.
- It seems like the diffusion community is moving more towards flow/ODE-based/consistency models, rather than DDPM-style models. Have the authors tried applying their method to flow-based methods?

---

> ### Author Response · Authors · 2024-11-21
>
> We appreciate the insightful feedback. We have made every effort to address your comments and revise the paper accordingly.
>
> -------
>
> ## **W1: Observation is already well-known and method is simple**
>
> First, we respectfully disagree with the reviewer’s opinion that the observation is already well-known. Previous studies (e.g., Karras et al., 2022; Ho et al., 2020; Hang et al., 2023) have offered conflicting perspectives on diffusion task difficulty, with some suggesting lower timesteps are more challenging and others suggesting the opposite. Our work brings clarity to this debate by analyzing task difficulty based on convergence speed and KL divergence, providing a grounded understanding that resolves these inconsistencies.
>
> Moreover, we would like to emphasize that the proposed method which, while simple, is based on novel observations and is not trivial. Naively applying curriculum learning can introduce noise due to variations in task difficulty, making task-wise clustering an essential component for mitigating these issues. Furthermore, based on our observation that convergence rates differ across curriculum phases, we designed a pacing function to dynamically adjust the training schedule. These two elements—task-wise clustering and the pacing function—work in tandem to create a robust framework that effectively enhances training stability and performance.
>
> Given these points, we believe our work provides a solid foundation for future research, and that the development of more sophisticated methods falls within the scope of future investigations. We kindly ask the reviewer to consider this point.
>
> -------
>
> ## **W2: Missing citations about training diffusion models from easy to difficult tasks**
>
> Progressive distillation [1] and cascaded diffusion [2] are fundamentally different from our approach, despite involving progressively more challenging tasks for the model. Progressive distillation focuses on reducing the number of sampling steps by training the model to progressively skip more steps, while cascaded diffusion aims to improve sample quality by progressively increasing the image resolution during training. Both methods concentrate on altering the model's behavior or structure to tackle specific challenges, such as efficiency or resolution enhancement.
> In contrast, our work identifies trends in task difficulty across timestep-wise denoising tasks and leverages these findings to propose an easy-to-hard training scheme. This training strategy directly addresses the order and structure of the learning process, optimizing task sequencing to enhance performance. This distinction emphasizes that our approach is fundamentally different from these methods, as it addresses a unique aspect of diffusion model training.
>
> -------
>
> ## **W3: Explanation of KL Divergence analysis**
>
> Thank you for pointing this out, and apologies for not explaining this more clearly. Your understanding is correct—higher KL divergence between the marginals of two timesteps implies that the model must make larger changes to the image. Furthermore, we note that the data distribution of $x_t$ becomes highly-peaked and narrow-supported as $t$ approaches 0, indicating that it is hard for the model to infer the $x_{t-1}$ with $x_t$. We will revise the text to clarify these points and ensure they are more clearly explained.
>
> -------
>
> ## **Q1: Explanation about improvement of anti-curriculum training**
>
> The performance improvement observed in anti-curriculum training seems to be due to the SNR-based clustering, rather than the hard-to-easy learning order. As investigated in (Go et al., 2023), SNR is closely related to task affinity, and clustering based on SNR ensures that tasks with similar noise levels are grouped together. This minimizes negative transfer and gradient conflicts that often arise in diffusion model training.
> As shown in Table 3, anti-curriculum combined with uniform clustering leads to worse performance than the vanilla method. However, when combined with SNR-based clustering, anti-curriculum training improves performance, suggesting that the way tasks are grouped plays a critical role in achieving optimal results.
>
> -------
>
> ## **Q2: Typo**
> Thank you for pointing this out. We will correct the typo and ensure the expression aligns with the results in Fig. 2.
>
> -------
>
> ## **Q3: Experiments on flow-based methods**
>
> We would like to highlight that we have experimented with a flow-based method, SiT, and validated the effectiveness of our proposed approach. As shown in Table 1, our method demonstrates consistent performance improvements, confirming its applicability to flow-based models as well.
>
> -------

---

> ### Comment · Reviewer_xMw1 · 2024-11-23
> **Thanks for the response!**
>
> I thank the authors for their detailed responses.
>
> **W1)** While it is clearly true that the observation that denoising difficulty changes with timestep is not novel, I agree that the paper provides a deeper and clearer analysis into this debate which is a useful contribution.
>
> On the topic of novelty, I saw that another reviewer raised [1] which was already cited in the paper, though in my opinion not sufficiently discussed. While [1] is not the same method (they adopt multi-task learning methods), it seems the core observations are similar, except that they use different terms ('affinities' instead of 'difficulties'). Can the authors clarify further the difference between their work and [1]? How are 'affinities' and 'difficulties' different? I think a deeper discussion in the paper will also help.
>
> **W2)** PD and CD are different works. I suggested citing them as I felt it will serve to contextualize the paper in the terms of the literature of easy-to-difficult task training in diffusion models, even if the tasks are different. This is not a major point so I leave this to the author's discretion.
>
> **W3/Q1-3)** I thank the authors for clarifications.
>
> I believe my current score is appropriate thus I leave it unchanged for now. However, I am curious about the author's opinions on [1] as if the similarities are unaddressed, it does impact the novelty of this work.
>
> [1] Go, Hyojun, et al. "Addressing negative transfer in diffusion models." Advances in Neural Information Processing Systems 36 (2024).

---

> > ### Author Response · Authors · 2024-11-23
> >
> > We sincerely appreciate your prompt feedback and the engaging discussion. We would like to address the points raised and clarify the differences and nuances highlighted.
> >
> > ---
> >
> > ### **W1) Regarding the relationship to [1] and the discussion of 'affinities' vs. 'difficulties':**
> >
> > We thank the reviewer for pointing out the need for a deeper discussion of [1]. While both our work and [1] explore the characteristics of denoising tasks in diffusion models, the aspects of exploration in each work are substantially different.
> >
> > The notion of task affinity introduced in [1, A] refers to how harmoniously the model can learn multiple tasks together. Specifically, their work focuses on identifying and mitigating conflicts between tasks, emphasizing task interactions and transferability by analyzing task similarities (e.g., gradient similarity or alignment).
> >
> > In contrast, our work explicitly quantifies the relative difficulty of individual denoising tasks across timesteps as a standalone property, independent of task interdependencies. The analysis of task difficulty in our work involves evaluating metrics such as loss behavior or convergence rates, directly reflecting the complexity of solving each task at different timesteps.
> >
> > Therefore, while [1] addresses how tasks relate and interact during multi-task learning, our focus lies in systematically characterizing the intrinsic difficulty of tasks across timesteps in diffusion models.
> >
> > [A] Efficiently Identifying Task Grouping for Multi-Task Learning, Neurips 2021.
> >
> > ---
> >
> > ### **W2) On citing PD and CD:**
> >
> > Thank you for the suggestion regarding PD and CD. We agree that their inclusion could provide useful context in terms of easy-to-difficult task training within diffusion models. We have revised it in Appendix A.2.

---

> > > ### Comment · Reviewer_xMw1 · 2024-11-27
> > > **Thanks for the response**
> > >
> > > Thanks for the clarification. I agree that there is a subtle difference between affinities and difficulties as defined by the authors which was not clear to me at first.
> > >
> > > I am satisfied with the response and believe my current score is appropriate.

---

> > > > ### Author Response · Authors · 2024-11-27
> > > >
> > > > Thank you for your thoughtful reply. We’re glad our clarification helped to address the subtle distinction between affinities and difficulties.
> > > >
> > > > We deeply appreciate the time and effort you’ve dedicated to reviewing our manuscript. Your invaluable suggestions and engagement in the discussion have significantly contributed to strengthening our submission.

---

### Official Review · Reviewer_Ujmn · 2024-10-29

**Soundness:** 2
**Presentation:** 3
**Contribution:** 2
**Rating:** 6
**Confidence:** 3

**Summary:**

This paper aims to improve the learning process of image generation using diffusion models through a technique called curriculum learning. The authors first observed the progress of diffusion model training. They found that denoising images with more noise (larger time steps) is easier than denoising images with less noise (smaller time steps). This was confirmed by examining the convergence of loss functions and FID scores at each noise level.

Furthermore, by analyzing the KL divergence between marginal probability distributions of consecutive time steps, they also demonstrated that the denoising task becomes more difficult with larger time steps.

Based on these observations, the authors adopted a common curriculum learning approach called "easy-to-hard training scheme." Specifically, they proposed a strategy that starts learning from time steps with more noise and gradually expands the range of time steps being learned. This strategy requires designing a "pacing function" to determine how to expand the learning range. The authors adopted a technique that transitions to the next phase based on the status of the training loss.

In experiments, they confirmed that the proposed method improves performance compared to vanilla learning strategies across multiple baseline models. Additionally, they demonstrated faster convergence and orthogonality (i.e., compatibility) with existing learning improvement methods.

**Strengths:**

The key features of this paper are as follows:

1. The motivation for the research is clearly explained in Section 4 (Observations). In particular, Figures 1 and 2 effectively illustrate the problems that need to be addressed.
2. The proposed method requires the design of a pacing function and scheduling, as described in Section 5.2. However, the core idea itself is simple and elegantly solves the identified problems.
3. In the experimental section, the method is tested on multiple baseline models, confirming that the proposed approach brings improvements in each case. Additionally, appropriate ablation studies have been conducted, suggesting the generalizability of the proposed method.

**Weaknesses:**

1. While the motivation in section 4 is clear, there is insufficient explanation as to why this method is expected to improve performance. Even though the "easy-to-hard training scheme" is well-known in the field of curriculum learning, the paper lacks discussion on why it works effectively and its connection to theoretical aspects.
2. It is appealing that the proposed method can generally improve upon baselines. However, from my understanding, the reported performance seems to be significantly different from the current state-of-the-art results. While achieving state-of-the-art performance is not mandatory for this type of paper, the lack of discussion about this performance gap raises concerns about the generalizability of the method.
3. This method presents a novel approach in applying curriculum learning to diffusion model training. However, there seems to be a lack of discussion regarding its relationship and comparison with other learning improvement techniques.

**Questions:**

1. Regarding Weakness 1: Can authors add theoretical justification or explanation, perhaps by citing other curriculum learning literature?
2. Also related to Weakness 1: Diffusion models differ from networks in other curriculum learning literatures in that the time step conditioning changes at each training phase. This might cause behaviors at different time steps (which should ideally be learned independently) to influence each other through shared network parameters. Therefore, I guess that the traditional curriculum learning framework might not fully explain the effectiveness of this method. Can you provide any references or discussion to address this question?
3. Regarding Weakness 2: Can you discuss why the performance of the proposed method is significantly inferior to current SoTA methods? For instance, the latest results on the ImageNet 256x256 dataset had FID scores below 5 as of 2022. (While additional experiments are not expected, if you could demonstrate improvement on a very high-performing model, it would strongly support the claims and effectiveness of this paper's method.)
4. Also related to Weakness 2: From Table 2, it seems that the results in Table 1 are from models that haven't fully been converged. While improved convergence performance is promising, it would be beneficial to show that performance improvements are still observed with further training for other models as well.
5. Regarding Weakness 3: Can you discuss the relationship between the proposed method and other diffusion model learning improvement techniques? For example, while this method is compatible with noise weighting techniques, I feel they might not be completely orthogonal in theory.
6. Can the KL divergence analysis in Section 4.2 be explicitly calculated in the forward process of diffusion?

Additional Notes
1. In the supplementary material, references are not properly cited.
2. When discussing convergence speed, for example in Figure 6, I believe it's important to include data points that show full convergence. Reaching the performance ceiling quickly is an advantage, so demonstrating this would be valuable.

---

> ### Author Response · Authors · 2024-11-21
>
> We are grateful to you for providing detailed and constructive comments, which are very helpful in improving our work. We will address all raised concerns by the reviewer and revise the paper accordingly.
>
> ---------
>
> ## **W1 & Q1: Lack of explanation about why easy-to-hard training is effective.**
> Thank you for your valuable feedback. While we briefly discussed the theoretical underpinnings of the easy-to-hard training paradigm (curriculum learning) in Section 2.3 of the related works—where we explained that curriculum learning starts from a smoother objective and gradually transforms into a less smooth version until it reaches the original objective function—we will provide a more detailed explanation with relevant references.
> Historically, there have been various theoretical explanations supporting the effectiveness of curriculum learning. For instance, (Bengio et al., 2009) introduced curriculum learning as a continuation method, starting with a smoother objective and gradually transitioning to a less smooth version until it reaches the original objective function. They demonstrated that this objective facilitates finding better local minima of a non-convex training criterion and accelerates convergence towards the global minimum. (Weinshall et al. 2018, Weinshall et al. 2020) analyzed curriculum learning in the context of convex optimization problems, such as linear regression loss and binary classification with hinge loss. Their findings demonstrated that curriculum learning significantly accelerates convergence speed, particularly during the initial training phase. By prioritizing simpler examples and gradually increasing complexity, this approach achieves faster optimization while maintaining robust training dynamics.(Saglietti et al. 2022) extended the understanding of curriculum learning using statistical physics methods in teacher-student networks. Their work highlighted how the careful selection of training examples based on difficulty can improve generalization performance and stabilize optimization, thereby contributing to the overall effectiveness of curriculum learning strategies.
> Since curriculum learning is widely recognized in the machine learning community, we did not delve deeply into its theoretical aspects and focused on analyzing the denoising tasks’ difficulties and proposing curriculum learning for diffusion model training based on the observation. However, we acknowledge the validity of the reviewer's comment and will allocate a section to thoroughly address the theoretical foundation and relevant references.
>
> ---------
>
> ## **W2 & Q3: Performance gap compared to state-of-the-art results**
>
> The performance differences observed are attributable to the specific experimental settings in our study. The results in Table 1 are based on DiT-L rather than DiT-XL, and the experiments in Table 4 for DiT-XL are limited to 400K training steps instead of the 7M steps typically required for state-of-the-art results. These choices were dictated by computational constraints.
>
> It is important to note that our experimental setup remains valid, as the DiT paper primarily conducted experiments using 400K iterations, which are distinct from the configurations used to achieve state-of-the-art results. Achieving such results typically necessitates significantly larger models and much longer training times, which were beyond the scope of our setup. Instead, we designed our experiments within a valid and practical framework, ensuring the reliability and relevance of our findings.
> To address your comment, we additionally conducted experiments on EDM2-S using the ImageNet-64 setup to ensure closer alignment with state-of-the-art results. For this experiment, we followed the default configuration provided in the official EDM2 repository, with the exception of the number of training iterations, which we limited to half due to time constraints during the rebuttal period. Under these conditions, the baseline EDM2-S achieved an FID of 1.97, while our method improved this further to an FID of 1.73. These results highlight the consistent performance gains achieved by our proposed approach, demonstrating its effectiveness and versatility even when applied to state-of-the-art methods. This further validates the robustness of our method across diverse experimental setups.
>
> ---------

---

> > ### Author Response · Authors · 2024-11-21
> >
> > ## **W3 & Q5: Lack of comparison with other diffusion training improvement techniques**
> > We would like to highlight that our method has already been demonstrated to be orthogonal to other advanced training techniques, such as architecture enhancements (DTR) and loss weighting (MinSNR). As shown in Table 5, the performance is significantly improved when our proposed curriculum learning is applied alongside these techniques.
> > In detail, MinSNR assigns loss weights to timesteps to prevent the model from focusing excessively on small noise levels. We have demonstrated that this loss weighting technique can positively complement our proposed easy-to-hard training approach, further enhancing its effectiveness.
> >
> > ---------
> >
> > ## **Q2: Discussion on applying curriculum learning in diffusion models**
> >
> > In multi-task learning (MTL) setups with shared parameters across tasks, prior research has demonstrated that curriculum learning can be highly effective. For instance, (Igarashi et al., 2022) introduced a curriculum learning approach for MTL based on gradient similarity. Their method prioritizes samples with fewer gradient conflicts during the early stages of training by assigning them higher weights. This approach not only reduces task interference but also improves overall performance, showing that parameter sharing in MTL is not a limitation but an opportunity for curriculum learning to resolve conflicts and enhance learning efficiency.
> > Building on these insights, the timestep-conditioned nature of diffusion models further supports their suitability for curriculum learning. Diffusion models inherently operate as MTL frameworks across timesteps, where each task corresponds to a denoising operation at a specific noise level, with shared parameters across these tasks (Go et al., 2023a). While the networks of diffusion models are conditioned on timesteps, the parameter-sharing mechanisms remain intact. Therefore, the application of curriculum learning is not hindered by this structure; rather, it seamlessly aligns with it, enabling diffusion models to benefit from reduced task interference and enhanced training efficiency.
> >
> > ---------
> >
> > ## **Q4: Longer training for other models**
> > Thank you for pointing this out. To address whether the performance improvements persist with further training for other models, we conducted additional experiments on SiT-B using the FFHQ dataset. As shown in the table below, the vanilla model converges after 250k iterations with FiD 5.65, our models achieve not only the same result faster at 200k iterations, but also outperforming results at 250k iterations with FiD 5.45. These results further validate the robustness of the proposed method across extended training durations.
> > |iterations(k)|       50|     100 |     150 |     200 |       250|     300 |     350 |     400 |     450 |     500 |
> > |------|---|---|---|---|---|---|---|---|---|---|
> > |SiT (Vanilla)|21.40|7.44|6.28|5.88|5.65|5.69|5.66|5.76|5.96|6.40|
> > |SiT + Ours|**14.38**|**6.95**|**6.00**|**5.63**|**5.45**|**5.46**|**5.45**|**5.53**|**5.69**|**6.14**|
> >
> > ---------
> >
> > ## **Q6: About KL divergence analysis**
> > Yes, the KL divergence was explicitly calculated in the forward process of diffusion. The calculations were performed using the actual data and the noise schedule. While we have provided these details in the supplementary material, we will include a more detailed explanation to enhance clarity and transparency.
> >
> > ---------
> >
> > ## **A1: References are missing in the supplementary material**
> > Apologies for the oversight. We appreciate your attention to detail and will ensure that the references are properly cited in the revised version.
> >
> > ---------
> >
> > ## **A2: Adding converged points when discussing convergence speed**
> > Thank you for constructive feedback. Please refer to Q4.
> >
> > ---------
> >
> > ## **References**
> > Igarashi et al., Multi-task Curriculum Learning Based on Gradient Similarity, BMVC 2022
> >
> > ---------

---

> > > ### Comment · Reviewer_Ujmn · 2024-11-26
> > >
> > > Thank you for the detailed rebuttal. I feel that most of my concerns have been addressed, but I would like to make one comment.
> > >
> > > In W3 & Q5, you claim in Table 5 that learning improvement techniques such as MinSNR and DTR are orthogonal to your proposed method. However, I believe that DTR+Ours does not significantly improve DTR. (While improvements in FID and IS are desirable, they are incremental compared to Vanilla.) Is it possible to discuss this matter? This is not meant as criticism, but rather as a way to discuss the relationship to existing methods.

---

> ### Author Response · Authors · 2024-11-26
>
> Thank you for your insightful feedback. It provided us with the opportunity to elaborate on the broader applicability of our method and clarify our rationale for selecting specific techniques. Below, we address your points in detail:
>
> ## **Discussion on Orthogonality with Existing Improvement Techniques**
> In fact, our method’s orthogonality with existing diffusion model improvement techniques has already been well demonstrated through our experiments even not in Table 5. To clarify, previous methods can be categorized as follows:
>
> **(1) Loss Weighting Techniques**
>
> These methods aim to improve diffusion model training by reweighting the loss function based on specific noise levels or tasks [A, B]. Examples can include MinSNR and noise weighting strategies in EDM and SiT. Among these, we selected MinSNR for our experiment to show orthogonality because it is a notable and widely recognized loss weighting strategy. More importantly, MinSNR is highly attachable to various diffusion models, which makes it an easy candidate for demonstrating orthogonality.
>
> **(2) Architectural Improvement Techniques**
>
> This category focuses on modifying the architecture of diffusion models, often introducing task-specific parameterizations or routing mechanisms. We chose DTR because it is easily attachable to various architectures. This allowed us to validate the compatibility of our method with architectural improvements in a straightforward and interpretable manner.
>
> **(3) Combined Techniques**
>
> Comprehensive approaches, such as EDM, EDM2 and SiT, integrate multiple improvement strategies, including noise scheduling, loss weighting, and architectural changes. Through our experiments, we verified that our method enhances performance even when applied to these combined methods, further demonstrating its broad compatibility and orthogonality.
> While the manuscript primarily focuses on MinSNR and DTR in Table 5, these were chosen deliberately because they are representative of simple, attachable techniques in their respective categories. We acknowledge that this choice may have understated the broader orthogonality demonstrated in conjunction with more complex frameworks like EDM and SiT. To address this, we will revise the manuscript to explicitly highlight the versatility of our method when applied to such combined techniques.
>
> ## **Modest Performance Gains in DTR + Ours**
>
> Regarding your observation about the incremental performance improvement in DTR + Ours compared to the vanilla baseline, we understand the concern. However, we view this result as an important validation of orthogonality. Specifically, it demonstrates that our method complements architectural improvements like DTR without interfering with their mechanisms.
> We do recognize that the term "significant improvement" might inadvertently overstate these results. To avoid any misinterpretation, we will revise the manuscript to focus on the consistent and complementary nature of our approach when combined with DTR, rather than emphasizing numerical gains alone.
>
> Thank you for your detailed and insightful feedback. While we have addressed the key points raised to the best of our ability in this revision, we acknowledge that a more thorough explanation of the relationships between our method and existing techniques, especially combined methods like EDM and SiT, could further strengthen the manuscript. Unfortunately, due to time constraints during the revision process (less than 1 day), we were unable to fully incorporate all these additional discussions.
> However, we deeply value your suggestions and are committed to reflecting these improvements in the future revision. Specifically, we will extend the discussion of orthogonality across diverse categories of improvement techniques, beyond what is currently presented, and provide more extensive experimental analyses where possible.
> Once again, thank you for your valuable comments, which have greatly contributed to improving the clarity and potential impact of our work.
>
> ## **References**
> [A] Perception Prioritized Training of Diffusion Models, CVPR 2022.
>
> [B] Addressing Negative Transfer in Diffusion Models, NeurIPS 2023.

---

> > ### Comment · Reviewer_Ujmn · 2024-11-27
> >
> > Thank you for your prompt response to the additional questions.
> >
> > Regarding the statement "our method complements architectural improvements like DTR without interfering with their mechanisms," I still have some doubts about whether this can truly be called "orthogonality." However, I agree that the characteristic of not hindering each other's improvement methods is indeed an important quality.
> >
> > Considering this, I would like to update my score for your work.

---

> > > ### Author Response · Authors · 2024-11-27
> > >
> > > We are glad to hear that you have increased the score for our paper. We deeply agree with your opinion that the orthogonality with DTR is not delivered through our current experimental results. In final version, we will do our best to supplement this claim.
> > >
> > > Thank you for your effort in reviewing our manuscript.

---

### Official Review · Reviewer_3pvQ · 2024-11-02

**Soundness:** 3
**Presentation:** 3
**Contribution:** 3
**Rating:** 6
**Confidence:** 4

**Summary:**

The paper proposes a curriculum learning based training of diffusion models, where the model is progressively trained on easier to harder tasks. The authors identify the task hardness based on the timestamps used for the training. For this, they train multiple models, where each model is trained to denoise only a particular subset of timestamp ranges. The loss curve of these models with training steps, depict that model trained on earlier timestamp ranges converges slower than the models trained on later timestamp ranges. Apart from the loss curves, the authors also look at the FID score by sampling from these models during training(to denoise or timestamps other than what the model is trained on, a model trained on all timestamps is used for denoising). Again the FID of models trained on later timestamps is lower than those trained on earlier timestamps.

Based on these analysis, the authors conclude that denoising for earlier timestamps is harder than denoising for later timestamps. Base on this, the authors then devise a curriculum learning-based training of diffusion models. The model is iteratively trained on later to earlier timestamps(easier to harder tasks) clusters. Instead of uniformly dividing the total timestamp into each cluster, the authors use an SNR-based interval clustering technique. Furthermore, since the hardness of different timestamp clusters varies, so the authors propose a better approach than training on each cluster for a fixed number of steps. They define a maximum threshold hyper-parameter(patience), which determines whether to switch to the next training cluster. If the loss in that cluster stays constant for more than patience steps, then the training proceeds to the next cluster.

The authors conduct experiments across different diffusion training architectures, such as as DiT, SiT and also over different training datasets such as FFHQ, ImageNet. Across different architectures and datasets, the proposed approach consistently outperforms the vanilla approach of training without a curriculum learning schedule.

**Strengths:**

* The authors show consistent improvement across a variety of architectures and datasets thus denoting the effectiveness of their approach.

* I like the anti-curriculum learning ablation, where instead of training on easier to harder tasks, the authors instead train on harder to easier tasks which doesn't perform any better than the baseline.

* I like the analysis done by the authors in determining the hardness of different timestamp schedules.

**Weaknesses:**

I have a few concerns regarding the paper -

* It seems that the effectiveness of the approach is reduced when the model is trained for longer steps. The difference in performance between the baseline and test after 2M training steps is much smaller than the difference at 400k steps. In that way, the main effectivess of the approach is just faster convergence, instead of improved performance. How do the authors justify the improved performance then?

* Can the authors show comparison between the curriculum and anti-curriculm approach for unconditional image generation also instead of just class-conditional generation?

* The authors should explain more in the paper about the SNR-based interval clustering technique and why it is a better interval technique than uniformly clustering.

**Questions:**

I have already asked questions in the weakness ection

---

> ### Author Response · Authors · 2024-11-21
>
> We appreciate your valuable review comments. We will address all your concerns and revise the paper accordingly.
>
> ---------
>
> ## **W1: Performance improvement on longer training**
>
> Thank you for pointing this out. As highlighted in Section 5.1 of the DiT paper, performance gains plateau beyond 2M iterations; for example, the DiT-XL/2 model reaches an FID of 2.55 at 2.35M iterations and only improves marginally to 2.27 after 7M iterations. In contrast, our proposed curriculum training method achieves significant improvements much earlier, as demonstrated in Table 2, where it outperforms the baseline at 2M iterations. This highlights that, under our method, 2M iterations can effectively be considered long training, as it accelerates convergence and delivers superior performance earlier in the process.
>
> ---------
>
> ## **W2: Comparison between the curriculum and anti-curriculum approach for unconditional image generation task**
>
> Thank you for your suggestion. We conducted additional experiments comparing the curriculum and anti-curriculum approaches for unconditional image generation using FFHQ with DiT-B. The results are as follows: curriculum (FID: 7.55) > vanilla (FID: 10.49) > anti-curriculum (FID: 15.53). These findings underscore the importance of the order in which training tasks are presented, as the curriculum approach consistently outperforms both the vanilla and anti-curriculum methods. This trend aligns with observations in class-conditional generation tasks, further highlighting the effectiveness of task ordering in achieving superior performance.
>
> ---------
>
> ## **W3: Explanation about SNR-based clustering**
> Thank you for pointing this out. We followed the SNR-based clustering approach used in (Go et al., 2023). The details are described in Section 4.1 of the paper. As highlighted in their work, SNR is closely related to task affinity, and clustering based on SNR ensures that tasks with similar noise levels are grouped together compared to uniform clustering.
> A well-structured task grouping is critical for curriculum learning. This aligns with findings from (Sarafianos et al., 2017), which demonstrated that grouping tasks with high affinity minimizes training conflicts and enhances learning efficiency. By leveraging SNR-based clustering, we establish intervals where tasks share strong affinities, resulting in a structured and smoother curriculum. This approach not only accelerates convergence but also improves final performance, as discussed earlier.
>
> ---------
>
> ## **Reference**
> Sarafianos et al., Curriculum Learning for Multi-Task Classification of Visual Attributes, ICCV 2017
>
> ---------

---

### Official Review · Reviewer_PdyQ · 2024-11-04

**Soundness:** 3
**Presentation:** 3
**Contribution:** 2
**Rating:** 6
**Confidence:** 4

**Summary:**

The paper notices that the diffusion training at low noise levels is more challenging than at high ones.
Based on this observation, the authors propose a curriculum learning approach for diffusion models (DMs) that facilitates faster convergence and improves overall performance.

**Strengths:**

**S1 |** The authors directly answer the question, "Which timestep intervals are more challenging during training?" and provide a convincing analysis for different diffusion parameterizations.

**S2 |** The experiments demonstrate that the proposed approach can improve the convergence and final performance of several popular DM methods.

**S3 |** The ablation study explores important questions, such as whether the performance gains persist with larger models and how the approach interacts with other training techniques, e.g., MinSNR loss weighting.

**Weaknesses:**

**W1 |** The proposed method has limited scientific contribution: the clustering is adopted from [1], the pacing function, while reasonable, is rather trivial, and the idea behind curriculum learning is pretty general. This could be fine if accompanied by very insightful and comprehensive analysis and strong results. Currently, I feel that the overall contribution is not sufficient.

**W2 |** Most experiments are performed using DiT, which currently seems to be a relatively weak baseline. EDM may also be considered outdated. I believe it is important to apply the proposed approach to EDM2[2] and demonstrate the gains on top of it. EDM2 focuses on training techniques and outperforms DiT and EDM by a large margin. Also, it proposes the dynamic loss weighting, which strongly relates to the proposed approach.

**W3 |** The analysis is performed only on FFHQ256 while the dataset and image resolution can be important factors as well. For example, [3] observed that larger models are more beneficial at high noise levels for CIFAR10 and ImageNet 64x64, and, in contrast, larger models are preferable at low noise levels for the LSUN dataset. [4] revealed different optimal timestep intervals for various datasets under the same noise schedule. Thus, it seems valuable to perform analyses across different datasets and discuss any observed trends. It would also be interesting to discuss whether pixel and latent spaces exhibit different behaviors.

---
[1] Go et al. Addressing Negative Transfer in Diffusion Models, 2023

[2] Karras et al. Analyzing and Improving the Training Dynamics of Diffusion Models, 2023

[3] Ganjdanesh et al. Mixture of Efficient Diffusion Experts Through Automatic Interval and Sub-Network Selection, 2024

[4] Liu et al. Oms-dpm: Optimizing the model schedule for diffusion probabilistic models, 2023

**Questions:**

**Q0 |** Please address the concerns and questions in Weaknesses.

**Q1 |** In Section 4.2, the relative entropy analysis indicates that marginal distributions become less similar as $t$ approaches 0. Could the authors elaborate on why this leads to the conclusion that training is more challenging at low noise levels? If I understand correctly, at small $t$, the tasks become more independent, which may limit the model to share knowledge between adjacent timesteps. Does this explanation seem reasonable?

**Q2 |** The method uses the SNR-based clustering. Did the authors consider using gradient-based clustering [1,2] in their approach? Perhaps, it may provide more accurate intervals for different datasets.

---

> ### Author Response · Authors · 2024-11-21
>
> We sincerely appreciate your valuable feedback on our paper. We have made every effort to address your comments and improve the manuscript accordingly.
>
> -----------
>
> ## **W1: Limited contribution**
> We respectfully disagree with the reviewer’s opinion that the contributions of our paper are insufficient. We believe our work provides meaningful insights and advancements for the diffusion modeling community. While it is true, as the reviewer points out, that curriculum learning is a general concept in machine learning and our method is relatively simple, we argue that our contributions extend beyond merely adopting existing ideas. Specifically, we:
> 1) Conduct an in-depth analysis of task difficulty in diffusion training, addressing an area with conflicting claims in prior studies.
> 2) Propose an easy-to-hard training scheme, which, while simple, is based on novel observations and is not trivial.
> 3) Thoroughly evaluate the proposed method through comprehensive experiments and ablation studies.
>
> Previous studies (e.g., Karras et al., 2022; Ho et al., 2020; Hang et al., 2023) have offered conflicting perspectives on diffusion task difficulty, with some suggesting lower timesteps are more challenging and others suggesting the opposite. Our work brings clarity to this debate by analyzing task difficulty based on convergence speed and KL divergence, providing a grounded understanding that resolves these inconsistencies.
>
> Building on this analysis, we propose an easy-to-hard training scheme that addresses key challenges in curriculum learning for diffusion training. Naively applying curriculum learning can introduce noise due to variations in task difficulty, making task-wise clustering an essential component for mitigating these issues as shown in our results. Furthermore, based on our observation that convergence rates differ across curriculum phases, we designed a pacing function to dynamically adjust the training schedule. These two elements—task-wise clustering and the pacing function—work in tandem to create a robust framework that effectively enhances training stability and performance.
>
> We validate the robustness and general applicability of our method across a wide range of models (e.g., DiT, SiT, EDM, EDM2) and datasets (e.g., FFHQ, ImageNet). Our experiments demonstrate that the proposed method remains effective with larger model sizes, longer training schedules, and advanced training techniques. Furthermore, we include extensive ablation studies to analyze the contribution of each component, improving understanding of the method.
> Given these points, we believe our work provides a solid foundation for future research, and that the development of more sophisticated methods falls within the scope of future investigations. We kindly ask the reviewer to consider this point.
>
> -----------
>
> ## **W2: Experiments on a stronger baseline, EDM2**
> Thank you for your valuable suggestion. To incorporate your comment, we additionally conducted experiments on EDM2-S on the ImageNet-64 setups to address your concern on EDM2. For the experiment, we followed the default configuration provided in the official EDM2 repository, except for the number of training iterations, which we limited to half due to time constraints during the rebuttal period. Under these conditions, the baseline EDM2-S achieved an FID of 1.97, while our method achieved an improved FID of 1.73. These results demonstrate that our proposed approach achieves consistent performance gains on top of EDM2, further validating its versatility even with state-of-the-art methods.
>
> -----------
>
> ## **W3: Analysis across different datasets and image resolution**
> Thank you for your constructive feedback. We would like to emphasize that our analyses have been conducted across not only different models but also different image resolutions: DiT (latent) and SiT (latent) with FFHQ256, and EDM (pixel) with FFHQ64. To further investigate the impact of datasets, we performed additional analyses on ImageNet256 using DiT, and observed consistent trends that the convergence speed of $M_i$ is faster as $i$ is larger as shown in Fig. B in our revised Supplementary Material. This suggests that our observations hold consistently across various datasets, resolutions, and diffusion spaces.
>
> -----------

---

> > ### Author Response · Authors · 2024-11-21
> >
> > ## **Q1: Explanation of KL divergence analysis**
> > Thank you for pointing this out, and apologies for not explaining this more clearly. Your understanding is correct—higher KL divergence between the marginals of two timesteps implies that the model must make larger changes to the image. Furthermore, we note that the data distribution of $x_t$ becomes highly-peaked and narrow-supported as $t$ approaches 0, indicating that it is hard for the model to infer the $x_{t-1}$ from  $x_t$. We will revise the text to clarify these points and ensure they are more clearly explained.
> >
> > -----------
> >
> > ## **Q2: Experiments using gradient-based clustering**
> > Thank you for pointing this out. We conducted experiments using gradient-based clustering as an alternative to SNR-based clustering. As shown in the table below, the results indicate that gradient-based clustering performs worse than both SNR-based and uniform clustering. This suggests that SNR-based clustering provides more effective intervals for our approach across different datasets. As shown in [1], each clustering method offers different performance trends, as the results do not direct the superiority of one clustering method. Therefore, the gradient-based clustering might not offer more accurate clustering results for denoising tasks.
> >
> > |Class-Conditional ImageNet 256x256.|       |      |      |      |
> > |------|---|---|---|---|
> > |*Curriculum Design*|*FID*|*IS*|*Prec*|*Rec*|
> > |Vanilla|30.27|60.06|0.55|0.52|
> > | + curriculum + uniform|25.01|71.99|0.58|**0.53**|
> > | + curriculum + SNR|**22.96**|**75.98**|**0.62**|0.52|
> > | + curriculum + Grad|26.72|70.34|0.58|0.52|
> >
> > -----------

---

> > > ### Comment · Reviewer_PdyQ · 2024-11-27
> > >
> > > I would like to thank the authors for their additional experiments and clarifications, as they have addressed most of my concerns. Based on this and the other responses, I have updated my score accordingly.

---

> > > > ### Author Response · Authors · 2024-11-27
> > > >
> > > > Thank you for your thoughtful feedback and for taking the time to review our additional experiments and clarifications. We are glad to hear that our responses have addressed most of your concerns. Your comments have been invaluable in helping us improve our work, and we truly appreciate your updated evaluation.

---

### Meta-Review · Area_Chair_PHrj · 2024-12-23

**Metareview:**

This paper provides a neat study of curriculum learning in diffusion models. The idea of training different diffusion models at different training regimes and analyzing the convergence to assess the task difficulty is quite interesting. This is an important contribution as there have been some debate on which noise region is easy to learn. The authors propose a way to cluster the noise regimes, show concretely with experiments that easy to hard curriculum learning helps. Experimental results are solid.

**Additional Comments On Reviewer Discussion:**

During the rebuttal, reviewers raised several concerns about lack of experiments, clarifications, etc. The authors sufficiently addressed these concerns. Post rebuttal, the paper looks in a good shape and all reviewers vote for accepting the paper. Hence, I think this is a good contribution for acceptance at ICLR.

---

### Decision · Program_Chairs · 2025-01-22

Accept (Poster)